# Kernel Metric Learning for In-Sample Off-Policy Evaluation of Deterministic RL Policies

**Haanvid Lee**[1]**, Tri Wahyu Guntara**[1]**, Jongmin Lee**[2]**, Yung-Kyun Noh**[3,4]**, Kee-Eung Kim**[1]
[1]KAIST, [2]UC Berkeley, [3]Hanyang Univ., [4]KIAS

## Abstract

We consider off-policy evaluation (OPE) of deterministic target policies for reinforcement learning (RL) in environments with continuous action spaces. While it is common to use importance sampling for OPE, it suffers from high variance when the behavior policy deviates significantly from the target policy. In order to address this issue, some recent works on OPE proposed in-sample learning with importance resampling. Yet, these approaches are not applicable to deterministic target policies for continuous action spaces. To address this limitation, we propose to relax the deterministic target policy using a kernel and learn the kernel metrics that minimize the overall mean squared error of the estimated temporal difference update vector of an action value function, where the action value function is used for policy evaluation. We derive the bias and variance of the estimation error due to this relaxation and provide analytic solutions for the optimal kernel metric. In empirical studies using various test domains, we show that the OPE with in-sample learning using the kernel with optimized metric achieves significantly improved accuracy than other baselines.

## 1 Introduction

Off-policy evaluation (OPE) aims to assess the performance of a target policy by using offline data sampled from a separate behavior policy, without the target policy interacting with the environment (Su et al., 2020; Fujimoto et al., 2021). OPE has gained considerable significance due to the potential costs and risks associated with a reinforcement learning (RL) agent interacting with real-world environments when evaluating a policy in domains such as healthcare (Zhao et al., 2009; Murphy et al., 2001), education (Mandel et al., 2014), robotics (Yu et al., 2020), and recommendation systems (Swaminathan et al., 2017). In the context of offline RL, OPE plays a vital role since the RL agent lacks access to the environment. Moreover, OPE can be leveraged to evaluate policies prior to their deployment in real-world settings and policy selection (Konyushova et al., 2021).

While OPE has been actively studied, OPE regarding deterministic target policies has not been extensively studied, and existing OPE algorithms either do not work or show low performance on evaluating deterministic policies (Schlegel et al., 2019; Fujimoto et al., 2021). However, there are many cases where deterministic policies are required. For example, safety-critical systems such as industrial robot control and drug prescription require precise and consistent control of action sequences without introducing variability (Silver et al., 2014; Kallus & Zhou, 2018). Therefore, the need for OPE of deterministic policies emerges. In this paper, we consider evaluating a deterministic policy with offline data.

Most of the recent works on off-policy evaluation (OPE) use marginalized importance sampling (MIS) approach (Xie et al., 2019; Nachum et al., 2019; Zhang et al., 2020). This approach learns stationary distribution correction ratios of states and actions, which are then utilized to reweight the rewards in the data. Since MIS methods use a single correction ratio, they have relatively lower variance than vanilla importance sampling (IS) methods, which use products of IS ratios of actions in trajectories (Precup et al., 2001; Levine et al., 2020). However, these methods are vulnerable to training instability due to optimizing the minimax objective. SR-DICE (Fujimoto et al., 2021) avoided optimizing the minimax objective by adopting a successor representation learning scheme. However, SR-DICE may use out-of-distribution samples in learning successor representation and thus may suffer from an extrapolation error.

In-sample learning methods learn Q-functions using only the samples in the data, avoiding extrapolation error (Schlegel et al., 2019; Zhang et al., 2023). The methods employ an IS ratio to correct the distribution of an action used for querying the Q-value in a temporal difference (TD) target to learn the Q-function. Since a single IS ratio is used, the method has lower variance than the traditional IS approaches (Levine et al., 2020; Precup et al., 2001). However, a limitation of these methods is their inapplicability to deterministic target policies, as the IS ratios of actions are almost surely zeros. To overcome this limitation, we aim to extend the use of IR to learn the Q-function in FQE (Voloshin et al., 2019; Le et al., 2019) style specifically tailored for deterministic target policies.

In this study, we introduce **K**ernel **M**etric learning for **I**n-sample **F**itted **Q** **E**valuation (KMIFQE), a novel approach that enables in-sample FQE using offline data. Our contribution is twofold: 1) enabling in-sample OPE of deterministic policies by kernel relaxation and metric learning that has not been used in MDP settings (Kallus & Zhou, 2018; Lee et al., 2022), 2) providing the theoretical guarantee of the proposed method. In theoretical studies, we first derive the MSE of our estimator with kernel relaxation applied on a target policy to avoid having zero IS ratios. From the MSE, we derive the optimal metric scale, referred to as bandwidth, which balances between bias and variance of the estimation and reduces the derived MSE. Then, we derive the optimal metric shape, referred to as metric, which minimizes the bias induced by the relaxation. Lastly, we show the error bound of the estimated Q-function compared to the true Q-function of a target policy. For empirical studies, we evaluate KMIFQE using a modified classic control domain sourced from OpenAI Gym (Brockman et al., 2016). This evaluation serves to verify that the metrics and bandwidths are learned as intended. Furthermore, we conduct experiments on a more complex MuJoCo domain (Todorov et al., 2012). The experimental results demonstrate the effectiveness of our metric learning approach.

## 2 RELATED WORK

**Marginalized importance sampling for OPE in RL** Marginalized importance sampling (MIS) OPE method was first introduced by Liu et al. (2018). They pointed out that the importance sampling (IS) OPE estimation in RL suffers from the "curse of horizon" due to products of IS ratios used for the IS. They avoided the problem by learning a correction ratio for the marginalized state distribution, which corrects the marginalized state distribution induced by a behavior policy to that of a target policy. Their work was limited in that it required a known behavior policy. Subsequent studies in MIS learned marginalized state-action distribution correction ratios and estimated policy values in a behavior-agnostic manner (Nachum et al., 2019; Zhang et al., 2020; Yang et al., 2020). However, the methods required optimization of minimax learning objectives, which caused instability in learning. SR-DICE (Fujimoto et al., 2021) avoided the minimax learning objective by using the concept of successor representation and made the learning process more stable. However, their result showed that FQE (Voloshin et al., 2019; Le et al., 2019) outperforms all MIS methods in a MuJoCo domain (Todorov et al., 2012) when the target policy is deterministic.

**In-sample learning for OPE in RL** In-sample learning methods (Schlegel et al., 2019; Zhang et al., 2023) learn a Q-function of a target policy by using the semi-gradients from expected SARSA temporal-difference (TD) learning loss computed on samples in the data. With importance resampling (IR), the methods correct the distribution of TD update vectors before updating the Q-function rather than reweighting the TD update vectors with IS ratios. They show that IR estimation of the TD update vector has a lower variance than the IS reweighted TD update vectors, resulting in more accurate Q-function estimation. However, these methods cannot be applied to the case where the target policy is deterministic since the IS ratios are zeros almost surely.

**Kernel metric learning for OPE in contextual bandits** The work of Kallus & Zhou (2018) first enabled IS to estimate the expected rewards for a deterministic target policy. They derived the MSE of the IS estimation when a kernel relaxes the target policy distribution. Then, they derived the optimal bandwidth (scale of the kernel metric) that minimizes it. Subsequent work by Lee et al. (2022) learned a shape of the kernel as a Mahalanobis distance metric (Mahalanobis, 1936; De Maesschalck et al., 2000) that further reduces MSE by reducing the bias of the estimate. These works cannot be directly applied to OPE in RL since that would result in IS estimations using products of IS ratios to correct the distributions of action sequences in trajectories sampled by a behavior policy. As the episode length becomes longer, more IS ratios are multiplied, and this causes the curse of horizon problem (Liu et al., 2018).

# 3 BACKGROUND

We seek to evaluate a deterministic target policy that operates in continuous action spaces given offline data sampled with a separate behavior policy. Specifically, we consider the MDP with $\mathcal{M} = \langle \mathcal{S}, \mathcal{A}, P, R, p_0, \gamma \rangle$ composed of a set of states $\mathcal{S} \in \mathbb{R}^q$, set of actions $\mathcal{S} \in \mathbb{R}^d$, reward function $R : \mathcal{S} \times \mathcal{A} \to \mathbb{R}$, state transition probability $P : \mathcal{S} \times \mathcal{A} \to \Delta(\mathcal{S})$, initial state distribution $p_0 \in \Delta(\mathcal{S})$, and a discount factor $\gamma \in [0, 1)$. We assume that the offline data $\mathcal{D} = \{(\mathbf{s}_i, \mathbf{a}_i, r_i, \mathbf{s}'_i, \mathbf{a}'_i)\}_{i=1}^n$ is sampled from the MDP $\mathcal{M}$ by using a stochastic policy $\mu : \mathcal{S} \to \Delta(\mathcal{A})$. And $\mathcal{D}$ is used to evaluate the deterministic target policy $\tilde{\pi} : \mathcal{S} \to \mathcal{A}$. The distribution of the target policy is denoted as $\pi : \mathcal{S} \to \Delta(\mathcal{A})$. The goal is to evaluate the target policy using the offline data $\mathcal{D}$ in a normalized discounted return $V(\pi) = (1 - \gamma)\mathbb{E}_\pi \left[ \sum_{t=0}^{\infty} \gamma^t R(\mathbf{s}_t, \mathbf{a}_t) \right]$, where $\mathbb{E}_\pi[\cdot]$ is the expectation using the data sampled from $\mathcal{M}$ with $\pi$. The target policy value can be represented with an Q-function $V(\pi) = (1 - \gamma)\mathbb{E}_{\mathbf{s}_0 \sim p_0(\mathbf{s}_0), \mathbf{a}_0 \sim \pi(\mathbf{a}_0 \mid \mathbf{s}_0)} [Q^\pi(\mathbf{s}_0, \mathbf{a}_0)]$, where the Q-function is defined as $Q^\pi(\mathbf{s}, \mathbf{a}) := \mathbb{E}_\pi \left[ \sum_{t=0}^{\infty} \gamma^t R(\mathbf{s}_t, \mathbf{a}_t) \mid \mathbf{s}_0 = \mathbf{s}, \mathbf{a}_0 = \mathbf{a} \right]$. This work estimates a Q-function to evaluate the target policy value.

## 3.1 IN-SAMPLE TD LEARNING

Temporal difference (TD) learning can be used for estimating $Q^\pi$ (Zhang et al., 2023). FQE learns a Q-function by minimizing the following TD loss $\mathcal{L}_{TD}(\theta)$ (Voloshin et al., 2019; Le et al., 2019):

$$\mathcal{L}_{TD}(\theta) = \mathbb{E}_{(\mathbf{s}, \mathbf{a}, \mathbf{s}') \sim p_\mu} \left[ \left( R(\mathbf{s}, \mathbf{a}) + \gamma \mathbb{E}_{\mathbf{a}' \sim \pi(\mathbf{a}' \mid \mathbf{s}')} \left[ Q_{\bar{\theta}}(\mathbf{s}', \mathbf{a}') \right] - Q_\theta(\mathbf{s}, \mathbf{a}) \right)^2 \right], \tag{1}$$

where $Q_\theta$ is Q-function parameterized by $\theta$, $Q_{\bar{\theta}}$ is a frozen target network, a transition data sampling distribution is defined as $p_\mu := d^\mu(\mathbf{s})\mu(\mathbf{a} \mid \mathbf{s})P(\mathbf{s}' \mid \mathbf{s}, \mathbf{a})p(r \mid \mathbf{s}, \mathbf{a})\mu(\mathbf{a}' \mid \mathbf{s}')$, and the stationary distribution of state induced by $\mu$ is:

$$d^\mu(\mathbf{s}) := (1 - \gamma) \sum_{t=0}^{\infty} \gamma^t p(\mathbf{s}_t = \mathbf{s} \mid \mathbf{s}_0 \sim p_0(\mathbf{s}_0), \mathbf{s}_t \sim P(\mathbf{s}_t \mid \mathbf{s}_{t-1}, \mathbf{a}_{t-1}), \mathbf{a}_t \sim \pi(\mathbf{a}_t \mid \mathbf{s}_t)). \tag{2}$$

In Eq. (1), $Q_\theta$ is trained only on $(\mathbf{s}, \mathbf{a}) \sim d^\mu(\mathbf{s})\mu(\mathbf{a} \mid \mathbf{s})$ while $Q_{\bar{\theta}}$ needs to be evaluated on out-of-distribution (OOD) samples $(\mathbf{s}', \mathbf{a}') \sim d^\mu(\mathbf{s}')\pi(\mathbf{a}' \mid \mathbf{s}')$ to compute the loss. Therefore, it may suffer from distributional shift and may produce inaccurate Q-values (Levine et al., 2020).

To avoid using OOD samples while computing the update vector for $Q_\theta$, gradient of $\mathcal{L}_{TD}(\theta)$ w.r.t. $\theta$, or the TD update vector $\Delta_{TD}$, can be represented with an IS ratio $w(\mathbf{s}, \mathbf{a}) := \frac{\pi(\mathbf{a} \mid \mathbf{s})}{\mu(\mathbf{a} \mid \mathbf{s})}$:

$$\Delta_{TD} = \mathbb{E}_{x \sim p_\mu} \Big[ w(\mathbf{s}', \mathbf{a}') \underbrace{\left( R(\mathbf{s}, \mathbf{a}) + \gamma Q_{\bar{\theta}}(\mathbf{s}', \mathbf{a}') - Q_\theta(\mathbf{s}, \mathbf{a}) \right) \nabla_\theta Q_\theta(\mathbf{s}, \mathbf{a})}_{=:\Delta(x)} \Big], \tag{3}$$

where a transition in the data is $x := \{\mathbf{s}, \mathbf{a}, r, \mathbf{s}', \mathbf{a}'\}$, $\Delta(x)$ is the semi-gradient of the transition $x$. By the importance ratio $w$, the data sampling distribution of $p_\mu$ is corrected to the distribution of $p_\pi := d^\mu(\mathbf{s})\mu(\mathbf{a} \mid \mathbf{s})P(\mathbf{s}' \mid \mathbf{s}, \mathbf{a})p(r \mid \mathbf{s}, \mathbf{a})\pi(\mathbf{a}' \mid \mathbf{s}')$.

In-sample TD learning (Schlegel et al., 2019; Zhang et al., 2023) uses IR to estimate $\Delta_{TD}$ as $\widehat{\Delta}_{IR}$:

$$\widehat{\Delta}_{IR} = \frac{\bar{w}}{k} \sum_{j=1}^{k} \Delta(\bar{x}_j), \quad \bar{x}_j \overset{\rho}{\sim} \{x_1, \ldots, x_n\} \text{ with probability } \rho_j = \frac{w(\mathbf{s}'_j, \mathbf{a}'_j)}{\sum_{i=1}^{n} w(\mathbf{s}'_i, \mathbf{a}'_i)}, \tag{4}$$

where $\bar{w} := \frac{1}{n} \sum_{i=1}^{n} w(\mathbf{s}'_i, \mathbf{a}'_i)$ is the bias correction term that corrects the bias induced by resampling from a finite data $\mathcal{D}$ with size $n$.

## 3.2 KERNEL METRIC LEARNING

As the IS ratios used for in-sample learning (Schlegel et al., 2019; Zhang et al., 2023) are almost surely zeros for a deterministic policy, it cannot be directly applied to the evaluation of a deterministic target policy. In this work, we enable in-sample learning to evaluate a deterministic target policy by relaxing the density of the target policy in an IS ratio, which can be seen as a Dirac delta function $\pi(\mathbf{a} \mid \mathbf{s}) = \delta(\mathbf{a} - \tilde{\pi}(\mathbf{s}))$, by a Gaussian kernel (Eq. (5)). We assume that the support of $\mu(\mathbf{a} \mid \mathbf{s})$ covers the support of $\frac{1}{h^d} K\left( \frac{L(\mathbf{s})^\top (\mathbf{a} - \tilde{\pi}(\mathbf{s}))}{h} \right)$ so the value of IS ratio is bounded. We seek to minimize the

MSE between the estimated TD update vector and the true TD update vector $\Delta_{TD}$ in Eq. (3) by learning Mahalanobis metrics $\frac{A(\mathbf{s})}{h^2}$ in their scales (hereafter referred to as bandwidths) $h$ and metrics in their shapes (hereafter referred to as metrics) $A(\mathbf{s})$ (Mahalanobis, 1936; De Maesschalck et al., 2000), which is locally learned at each state $\mathbf{s}$ (Eq. (6)). The metrics are locally learned at each state $\mathbf{s}$ to reflect the Q-value landscape of the target policy near the target actions $\tilde{\pi}(\mathbf{s})$.

$$w(\mathbf{s}, \mathbf{a}) = \frac{\delta(\mathbf{a} - \tilde{\pi}(\mathbf{s}))}{\mu(\mathbf{a} \mid \mathbf{s})} \approx \frac{1}{h^d \mu(\mathbf{a} \mid \mathbf{s})} K\left(\frac{L(\mathbf{s})^\top (\mathbf{a} - \tilde{\pi}(\mathbf{s}))}{h}\right), \tag{5}$$

$$\frac{1}{h^d} K\left(\frac{L(\mathbf{s})^\top (\mathbf{a} - \tilde{\pi}(\mathbf{s}))}{h}\right) = \frac{1}{h^d (2\pi)^{\frac{d}{2}}} \exp\left(-\frac{(\mathbf{a} - \tilde{\pi}(\mathbf{s}))^\top A(\mathbf{s})(\mathbf{a} - \tilde{\pi}(\mathbf{s}))}{h^2}\right), \tag{6}$$

where we assume that $|A(\mathbf{s})| = 1, A(\mathbf{s}) \succ 0, A(\mathbf{s})^\top = A(\mathbf{s})$. Applying Mahalanobis metric $(A(\mathbf{s}) = L(\mathbf{s})L(\mathbf{s})^\top)$ to a Gaussian kernels can be viewed as linearly transforming the inputs with the transformation matrix $L(\mathbf{s})$ as in Eq. (6) (Noh et al., 2010; 2017).

## 4 KERNEL METRIC LEARNING FOR IN-SAMPLE TD LEARNING

Our work enables the in-sample estimation of TD update vector $\widehat{\Delta}_{IR}^K$ with kernel relaxation and metric learning for estimating $Q^\pi$ which is used to evaluate a deterministic target policy. The in-sample estimation is enabled by relaxing the density of a deterministic target policy in an importance sampling (IS) ratio in Eq. (4) by a Gaussian kernel as in Eq. (5).

$$\widehat{\Delta}_{IR}^K = \frac{\bar{w}^K}{k} \sum_{j=1}^k \Delta\left(\bar{x}_j\right), \quad \bar{x}_j \overset{\rho_j^K}{\sim} x_1, \ldots, x_n \text{ with probability } \rho_j^K = \frac{w^K(\mathbf{s}_j', \mathbf{a}_j')}{\sum_{i=1}^n w^K(\mathbf{s}_i', \mathbf{a}_i')}, \tag{7}$$

where $w^K(\mathbf{s}_i', \mathbf{a}_i') := \frac{1}{h^d \mu(\mathbf{a}_i' \mid \mathbf{s}_i')} K\left(\frac{L(\mathbf{s}_i')^\top (\mathbf{a}_i' - \tilde{\pi}(\mathbf{s}_i'))}{h}\right)$, and the bias correction term with the kernel relaxation is denoted as $\bar{w}^K := \frac{1}{n} \sum_{i=1}^n w^K(\mathbf{s}_i', \mathbf{a}_i')$. The relaxation reduces variance but increases bias. Next, we analyze the bias and variance that together compose MSE of the proposed estimate and aim to find kernel bandwidths and metrics that best balance the bias and the variance to minimize the MSE of a TD update vector estimation.

### 4.1 MSE DERIVATION

To derive the MSE of the estimated TD update vector $\widehat{\Delta}_{IR}^K$ while assuming an isotropic kernel $(A(\mathbf{s}') = L(\mathbf{s}') = I)$ is used, The bias and variance of $\widehat{\Delta}_{IR}^K$ are derived.

**Assumption 1.** *Support of the behavior policy $\mu(\mathbf{a} \mid \mathbf{s})$ contains the actions determined by the target policy $\tilde{\pi}(\mathbf{s})$ and the support of its kernel relaxation $\frac{1}{h^d} K\left(\frac{L(\mathbf{s})^\top (\mathbf{a} - \tilde{\pi}(\mathbf{s}))}{h}\right)$.*

**Assumption 2.** *The target Q network $Q_{\bar{\theta}}(\mathbf{s}, \mathbf{a})$ is twice differentiable w.r.t. an action $\mathbf{a}$.*

**Theorem 1.** *Under Assumption 1-2, the bias and variance of $\widehat{\Delta}_{IR}^K$ are:*

$$\text{Bias}[\widehat{\Delta}_{IR}^K] = h^2 \underbrace{\frac{\gamma}{2} \mathbb{E}_{\mathcal{D} \sim p_\mu} \left[\nabla_{\mathbf{a}'}^2 Q_{\bar{\theta}}(\mathbf{s}', \mathbf{a}')|_{\mathbf{a}' = \tilde{\pi}(\mathbf{s}')} \nabla_\theta Q_\theta(\mathbf{s}, \mathbf{a})\right]}_{=:\mathbf{b}} + \mathcal{O}(h^4), \tag{8}$$

$$\text{Var}[\widehat{\Delta}_{IR}^K] = \frac{1}{nh^d} \underbrace{C(K) \mathbb{E}_{p_\mu} \left[\frac{(r + \gamma Q_{\bar{\theta}}(\mathbf{s}', \tilde{\pi}(\mathbf{s}')) - Q_\theta(\mathbf{s}, \mathbf{a}))^2 \|\nabla_\theta Q_\theta(\mathbf{s}, \mathbf{a})\|_2^2}{\mu(\tilde{\pi}(\mathbf{s}') \mid \mathbf{s}')}\right]}_{=:v} + \mathcal{O}\left(\frac{1}{nh^{d-2}}\right), \tag{9}$$

*where $\nabla_{\mathbf{a}'}^2$ is a Laplacian operator w.r.t. $\mathbf{a}'$, $\text{Var}[\mathbf{z}] := \text{tr}[\text{Cov}(\mathbf{z}, \mathbf{z})]$ for a vector $\mathbf{z}$, $\mathbf{b}$ is the bias constant vector, $v$ is the variance constant, $C(K) := \int K(\mathbf{z})^2 d\mathbf{z}$, $h^2 \mathbf{b}$ is the leading-order bias, and $\frac{v}{nh^d}$ is the leading-order variance.*

In Theorem 1, as the bandwidth $h$ increases, the leading-order bias increases, and the leading-order variance decreases as a bias-variance trade-off. For the leading-order variance, it increases when the L2 norm of the TD update vector ($\Delta(x)$ in Eq. (9) with $x = \{\mathbf{s}, \mathbf{a}, r, \mathbf{s}', \tilde{\pi}(\mathbf{s}')\}$) is large, and this is understandable since the variance of a random variable (estimated TD update vector) would be increased if the scale of the random variable is increased. The leading-order bias increases when the

second-order derivative of Q-function is large. This is reasonable since the estimation bias would increase when the difference between $Q_{\bar{\theta}}(\mathbf{s}', \mathbf{a}')$ and $Q_{\bar{\theta}}(\mathbf{s}', \tilde{\pi}(\mathbf{s}'))$ grows.

The bias and variance of the TD update vector estimation $\widehat{\Delta}_{IR}^{K}$ are derived by applying the law of total expectation and variance since the samples used for estimating $\widehat{\Delta}_{IR}^{K}$ are first sampled with $p_{\mu}$ and then resampled with $\rho^{K}$. Taylor expansion is also used for the derivation on $\mu(\mathbf{a}' \mid \mathbf{s}')$ and target network $Q_{\bar{\theta}}(\mathbf{s}', \mathbf{a}')$ at $\mathbf{a}' = \tilde{\pi}(\mathbf{s}')$ and the odd terms of Taylor expansion are canceled out due to the symmetric property of $K$ (proof in Appendix A.1).

Our derivation of bias and variance differs from the kernel metric learning methods for OPE in contextual bandits (Kallus & Zhou, 2018; Lee et al., 2022) in that it is the bias and variance of the update vector of the OPE estimate rather than the bias and variance of the OPE estimate. The derivation also differs from the derivations in previous in-sample learning methods (Schlegel et al., 2019; Zhang et al., 2023) in that we derive the bias in terms of $h, n, d$ to analyze the bias and variance w.r.t. these variables while the derivations in the previous methods are made to mainly compare bias and variance of IR and IS estimates.

From the bias and variance in Theorem 1, MSE can be derived (proof in Appendix A.2).

**Corollary 1.** *Under Assumption 1-2, MSE between the TD update vector estimate $\widehat{\Delta}_{IR}^{K}$ and the on-policy TD update vector $\Delta_{TD}$ is:*

$$\mathrm{MSE}(h, n, k, d) = \underbrace{h^4 \| \mathbf{b} \|_2^2 + \frac{v}{nh^d}}_{=:\mathrm{LOMSE}(h,n,d)} + \mathcal{O}\left(h^6\right) + \mathcal{O}\left(\frac{1}{nh^{d-2}}\right), \tag{10}$$

*where* $\mathrm{LOMSE}(h, n, d)$ *is the leading-order MSE.*

## 4.2 OPTIMAL BANDWIDTH AND METRIC

This section derives the optimal bandwidth and metric from the leading order MSE (LOMSE) in Eq. (10). The optimal bandwidth should minimize the LOMSE by balancing the bias-variance trade-off.

**Proposition 1.** *The optimal bandwidth $h^*$ that minimizes the $\mathrm{LOMSE}(h, n, d)$ is:*

$$h^* = \left(\frac{vd}{4n\| \mathbf{b} \|_2^2}\right)^{\frac{1}{d+4}}. \tag{11}$$

The optimal bandwidth $h^*$ is derived by taking the derivative on $\mathrm{LOMSE}(h, n, d)$ w.r.t. the bandwidth $h$ similarly to the work of Kallus & Zhou (2018) since $\mathrm{LOMSE}(h, n, d)$ is convex w.r.t. $h$ when $h > 0$ (derivation in Appendix A). In Eq. (11), notice that the optimal bandwidth $h^* \to 0$ as $n \to \infty$ and LOMSE becomes dominant in MSE. Furthermore, the $h^*$ increases when the variance constant $v$ is large, and the L2 norm of the bias constant vector $\mathbf{b}$ is small, and vice versa, to balance between bias and variance and minimize LOMSE (derivation in Appendix A.3).

Next, we analyze how the LOMSE is affected by $d$ when $h^*$ is applied to our algorithm.

**Proposition 2.** *Given optimal bandwidth $h^*$ (Eq. (11)), in high-dimensional action space, the squared L2-norm of leading-order bias $h^4 \| \mathbf{b} \|_2^2$ dominates over the leading-order variance $\frac{v}{nh^d}$ in LOMSE (Eq. (10)). Furthermore, $\mathrm{LOMSE}(h^*, n, D)$ approximates to $\| \mathbf{b} \|_2^2$.*

The proof involves plugging in the optimal bandwidth $h^*$ in Eq. (11) to the LOMSE in Eq. (10) and taking the limit $d \to \infty$ (derivation in Appendix A.4). Proposition 2 suggests that the MSE of the estimate $\widehat{\Delta}_{IR}^{K}$ is dominated by the bias in high-dimensional action space when the optimal bandwidth $h^*$ is applied. Therefore, in a high-dimensional action space, a significant amount of MSE can be reduced by reducing the bias. We seek to further minimize the bias that dominates the MSE in high-dimensional action space with metric learning.

Since applying the Mahalanobis metric $A(\mathbf{s}) = L(\mathbf{s})L(\mathbf{s})^{\top}$ to a kernel is equivalent to linearly transforming the kernel inputs with the linear transformation matrix $L(\mathbf{s})$ (Section 3.2), we analyze the effect of the kernel metric on the bias constant vector $\mathbf{b}$ with the linearly transformed kernel inputs. The squared L2-norm of the bias constant vector with kernel metric is:

$$\|\mathbf{b}_A\|_2^2 = \frac{\gamma^2}{4} \|\mathbb{E}_{\mathcal{D} \sim p_{\mu}} \left[\mathrm{tr}\left[A(\mathbf{s}')^{-1} \mathbf{H}_{\mathbf{a}'} Q_{\bar{\theta}}(\mathbf{s}', \mathbf{a}')|_{\mathbf{a}'=\tilde{\pi}(\mathbf{s}')}\right] \nabla_{\theta} Q(\mathbf{s}, \mathbf{a})\right]\|_2^2, \tag{12}$$

where $\mathbf{H}_{\mathbf{a}'}$ is the Hessian operator w.r.t. $\mathbf{a}'$. The derivation involves a change-of-variable on conditional distribution and derivatives (derivation in Appendix A.5).

Minimizing Eq. (12) is challenging as it necessitates a comprehensive examination of the collective impact exerted by every metric matrix $A(\mathbf{s}')$ inside the expectation. Therefore, an alternative approach is adopted wherein the focus shifts towards minimizing the subsequent upper bound $U(A)$ in Eq. (13). This alternative strategy enables the derivation of a closed-form metric matrix for each state, employing a nonparametric methodology while being bandwidth-agnostic.

$$
\min_{\substack{A:\ A(\mathbf{s}')\succ 0, \\ A(\mathbf{s}')=A(\mathbf{s}')^\top, \\ |A(\mathbf{s}')|=1\ \forall \mathbf{s}'}} U(A) = \frac{\gamma^2}{4}\mathbb{E}_{\mathcal{D}\sim p_\mu}\left[\mathrm{tr}\left(A(\mathbf{s}')^{-1}\,\mathbf{H}_{\mathbf{a}'}\,Q_{\bar{\theta}}(\mathbf{s}',\mathbf{a}')\big|_{\mathbf{a}'=\tilde{\pi}(\mathbf{s}')}\right)^2 \|\nabla_\theta Q(\mathbf{s},\mathbf{a})\|_2^2\right]. \quad (13)
$$

**Proposition 3.** *Under Assumption 2, define $\Lambda_+(\mathbf{s}') \in \mathbb{R}^{d_+(\mathbf{s}')\times d_+(\mathbf{s}')}$ and $\Lambda_-(\mathbf{s}') \in \mathbb{R}^{d_-(\mathbf{s}')\times d_-(\mathbf{s}')}$ as diagonal matrices of positive and negative eigenvalues from the Hessian $\mathbf{H}_{\mathbf{a}'}Q_{\bar{\theta}}(\mathbf{s}',\mathbf{a}')|_{\mathbf{a}'=\tilde{\pi}(\mathbf{s}')}$, and define $U_+(\mathbf{s}')$ and $U_-(\mathbf{s}')$ as the matrices of eigenvectors corresponding to $\Lambda_+(\mathbf{s}')$ and $\Lambda_-(\mathbf{s}')$ respectively. Then the $U(A)$ minimizing metric $A^*(\mathbf{s}')$ is:*

$$
A^*(\mathbf{s}') = \alpha(\mathbf{s}')\left[U_+(\mathbf{s}')U_-(\mathbf{s}')\right]\underbrace{\begin{pmatrix} d_+(\mathbf{s}')\Lambda_+(\mathbf{s}') & 0 \\ 0 & -d_-(\mathbf{s}')\Lambda_-(\mathbf{s}') \end{pmatrix}}_{=:M(\mathbf{s}')}\left[U_+(\mathbf{s}')U_-(\mathbf{s}')\right]^\top, \quad (14)
$$

*where $\alpha(\mathbf{s}') := |M(\mathbf{s}')|^{-1/\left(d_+(\mathbf{s}')+d_-(\mathbf{s}')\right)}$.*

The proof entails solving Eq. (13) for $A(\mathbf{s}')$ for each $\mathbf{s}'$ (thus locally learn metrics for each $\mathbf{s}'$) by means of the Lagrangian equation. The closed-form solution to the minimization objective involving squared trace term of two matrix products in Eq. (13) is reported in the work of Noh et al. (2010) that learns kernel metric for kernel regression, and the solution had been used in the context of kernel metric learning for OPE in contextual bandits (Lee et al., 2022). We use the same solution for acquiring kernel metrics that minimize the squared trace term.

Notice that when the Hessian $\mathbf{H}_{\mathbf{a}'}\,Q_{\bar{\theta}}(\mathbf{s}',\mathbf{a}')|_{\mathbf{a}'=\tilde{\pi}(\mathbf{s}')}$ contains both positive and negative eigenvalues, the trace in Eq. (13) becomes zero. The optimal metric $A^*(\mathbf{s}')$ regards the next action $\mathbf{a}'$ in data is similar to the next target action $\tilde{\pi}(\mathbf{s}')$ when the eigenvalue of the Hessian in that direction is small, and decreases the Mahalanobis distance between $\mathbf{a}'$ and $\tilde{\pi}(\mathbf{s}')$. Consequently, as the Mahalanobis distance decreases, kernel value in the IS ratio $w^K(\mathbf{s}',\mathbf{a}')$ increases. Finally, the associated resampling probability increases, leading to a higher resampling probability.

Utilizing the derived optimal bandwidth $h^*$ and optimal metric $A^*$, the TD update vector for $Q_\theta$ can be estimated in an in-sample learning manner by using importance resampling (IR). The estimated TD update vector is then used for learning $Q_\theta$. As $Q_\theta$ is learned by bootstrapping, the actual algorithm iterates between learning $h^*$, $A^*$, and learning $Q_\theta$. Finally, when $Q_\theta$ converges, it is used for evaluating the deterministic target policy. The detailed procedure is in Algorithm 1 in Appendix B. Although we assumed a known behavior policy, our method can be applied to offline data that is sampled from unknown multiple behavior policies by estimating a behavior policy by maximum likelihood estimation. The work by Hanna et al. (2019) shows that the IS estimation of a target policy value is more accurate when an MLE behavior policy is used instead of the true behavior policy.

## 4.3 Error Bound Analysis

In this section, we analyze the error bound of the estimated Q-function with KMIFQE. The difference between the on-policy TD update vector $\Delta_{TD}$ and the KMIFQE estimated TD update vector $\widehat{\Delta}_{IR}^K$ is that $\widehat{\Delta}_{IR}^K$ is obtained from the TD loss in Eq. (1) with $K$ instead of $\pi$. In other words, updating $\theta$ using $\widehat{\Delta}_{IR}^K$ can be understood as evaluating the *stochastic* target policy relaxed by the bandwidth $h$ and the matric $A$. We will show the evaluation gap resulting from using the relaxed stochastic policy instead of a deterministic policy. To this end, we first define Bellman operators $T$ and $T_K$ for the deterministic target policy $\tilde{\pi}$ and the stochastic target policy $\pi_K(\mathbf{a}\,|\,\mathbf{s}) = \frac{1}{h^d}K\left(\frac{L(\mathbf{s})^\top(\mathbf{a}-\tilde{\pi}(\mathbf{s}))}{h}\right)$ respectively.

$$
TQ(\mathbf{s},\mathbf{a}) := R(\mathbf{s},\mathbf{a}) + \gamma\mathbb{E}_{\mathbf{s}'\sim P(\mathbf{s}'|\mathbf{s},\mathbf{a})}\left[Q\left(\mathbf{s}',\tilde{\pi}\left(\mathbf{s}'\right)\right)\right], \quad (15)
$$

$$
T_KQ(\mathbf{s},\mathbf{a}) := R(\mathbf{s},\mathbf{a}) + \gamma\mathbb{E}_{\mathbf{s}'\sim P(\mathbf{s}'|\mathbf{s},\mathbf{a}),\mathbf{a}'\sim\pi_K(\mathbf{a}'\,|\,\mathbf{s}')}\left[Q\left(\mathbf{s}',\mathbf{a}'\right)\right]. \quad (16)
$$

We then analyze the difference in fixed points for each Bellman operator.

**Assumption 3.** *For all $\{\mathbf{s}, \mathbf{a}\} \in S \times A$, assume that the second-order derivatives of an arbitrary $Q(\mathbf{s}, \mathbf{a})$ w.r.t. actions are bounded.*

**Theorem 2.** *Under Assumption 3, denoting $m$ iterative applications of $T$ and $T_K$ to an arbitrary $Q$ as $T^m Q$ and $T_K^m Q$ respectively, and $h$ and $A(\mathbf{s}')$ as an arbitrary bandwidth and metric respectively, the following holds:*

$$\|Q^\pi - \lim_{m \to \infty} T_K^m Q\|_\infty \le \frac{\gamma \xi}{1 - \gamma}, \tag{17}$$

$$\xi := \max_{m, \{\mathbf{s}, \mathbf{a}\} \in S \times A} \frac{h^2}{2} \mathbb{E}_{\mathbf{s}' \sim P(\mathbf{s}' \mid \mathbf{s}, \mathbf{a})} \left[ \left| \text{tr} \left( A(\mathbf{s}')^{-1} \mathbf{H}_{\mathbf{a}'} T_K^m Q(\mathbf{s}', \mathbf{a}')|_{\mathbf{a}' = \tilde{\pi}(\mathbf{s}')} \right) \right| \right] + |\mathcal{O}(h^4)|. \tag{18}$$

The proof involves deriving $\xi$ that upperbounds $\|TQ - T_K Q\|_\infty / \gamma$ by using Taylor expansion at $\mathbf{a}' = \tilde{\pi}(\mathbf{s}')$, and showing $\|T^\infty Q - T_K^\infty Q\|_\infty \le \sum_{i=1}^\infty \gamma^i \xi$ using mathematical induction (proof in Appendix A.6).

Theorem 2 shows that the evaluation gap is bounded by the degree of stochastic relaxation $h$ and metric metric $A(\mathbf{s}')$. It is noteworthy that for any given $h$, we can always reduce the gap by applying optimal metric $A^*(\mathbf{s}')$ in Proposition 3, which will also be shown empirically in the experiments (Figure 1b). While $h = 0$ seems to be always preferred in Theorem 2, it is because it only considers the exact policy evaluation without any estimation error by using finite samples. When we update $Q$ using finite samples, the bias-variance trade-off by choice of $h$ should be considered, as shown in Theorem 1.

## 5 EXPERIMENTS

In this section, we empirically validate the theoretical findings related to the learned metrics and bandwidths in Section 4.2 and compare the performance of KMIFQE with the baselines. The baselines include SR-DICE (Fujimoto et al., 2021) and FQE (Voloshin et al., 2019), which are state-of-the-art model-free OPE algorithms for evaluating deterministic target policies (Fujimoto et al., 2021) but vulnerable to extrapolation errors due to the usage of OOD samples. KMIFQE is evaluated on three test domains. First, we evaluate KMIFQE on OpenAI gym Pendulum-v0 environment (Brockman et al., 2016) with dummy action dimensions to see if the metrics and bandwidths are learned as intended. Then, we compare KMIFQE with the baselines on more complex MuJoCo environments (Todorov et al., 2012). Lastly, KMIFQE and baselines are evaluated on D4RL (Fu et al., 2020) datasets sampled from unknown multiple behavior policies since the baselines assume unknown multiple behavior policies. To apply KMIFQE, which assumes a known behavior policy, on the D4RL datasets, maximum likelihood estimated (MLE) behavior policies are used. For the hyperparameters in the baselines, we used the hyperparameters in the work of Fujimoto et al. (2021) as our test data is the same or similar to theirs. For KMIFQE, we used the same hyperparameter settings to the baselines for those overlap and IS ratio clipping in the range of [1e-3, 2] selected by grid search (see Appnedix C).

### 5.1 PENDULUM WITH DUMMY ACTION DIMENSIONS

Pendulum-v0 environment (Brockman et al., 2016) is modified by adding dummy action dimensions that are unrelated to the estimation of target policy values for imposing large OPE estimation bias when an OPE algorithm fails to ignore the dummy action dimensions. The domain is prepared to observe if the KMIFQE metrics are learned to reduce the bias of $\widehat{\Delta}_{IR}^K$ as intended in Proposition 3, leading to less error in the estimation of $Q^\pi$ as stated in Theorem 2. For the dataset, actions in the dummy action dimensions are uniformly sampled from the original action range $[-a_{\max}, a_{\max}]$, and the original action is sampled from the mixture of 20% uniform random distribution and 80% Gaussian density of $N(\tilde{\mu}_1(\mathbf{s}), (0.5a_{\max})^2)$ where $\tilde{\mu}_1$ is a TD3 policy showing medium-level performance. The target policy $\tilde{\pi}(\mathbf{s})$ outputs action of $\tilde{\pi}_1(\mathbf{s})$ for the original action dimension and zeros for the dummy action dimensions. $\tilde{\pi}_1(\mathbf{s})$ is a TD3 policy showing expert-level performance. The dataset contains 0.5 million transitions (more details in Appendix C).

We examine if our theoretical analysis made in Section 4.2 is supported by empirical results. Due to the unavailability of the ground truth TD update vectors, we analyze the impact of metric and bandwidth learning on the MSE of the estimated target policy values instead of evaluating its effect on the estimated TD update vector. We hypothesize that an estimation error in the TD update vector would lead to an estimation error in the target policy values.

Firstly, Proposition 2 is empirically validated in Figure 1a. In the figure, empirical bias becomes much larger than the variance with KMIFQE learned bandwidth as the number of dummy action dimensions increases. Furthermore, the bias is reduced by the KMIFQE learned metrics (Proposition 3). Secondly, in Figure 1b, we show that the learned metrics reduce biases in all bandwidths as claimed in Proposition 3. Furthermore, the "$U$" shaped curve shows the bias-variance trade-off on varying bandwidths. Notably, the learned bandwidth balances between bias and variance as intended in Proposition 1. Finally, KMIFQE learned metrics and Q-value landscapes are presented in Figure 2 showing that the metrics are learned to ignore the dummy action dimensions and the estimated Q-value landscape changes a little along the dummy action dimension compared to the one estimated by FQE. Moreover, the KMIFQE correctly estimates high Q-values for target actions that are optimal, while FQE estimates low Q-values for the actions.

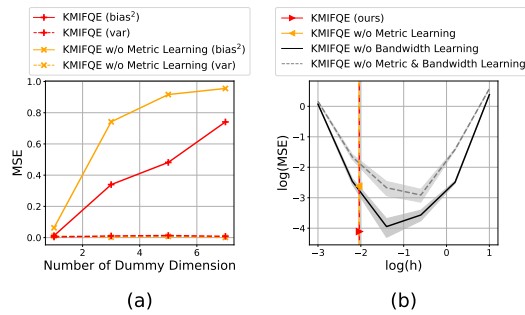

Figure 1: (a) Empirical bias and variance of the KMIFQE with and without metric learning as the number of dummy action dimensions increases. (b) Performance of KMIFQE with and without metric learning under various given bandwidths. The bandwidths learned by KMIFQE are plotted as vertical lines along with makers indicating the MSEs. The shaded area is the region within one standard error. All experiments are repeated for 10 trials.

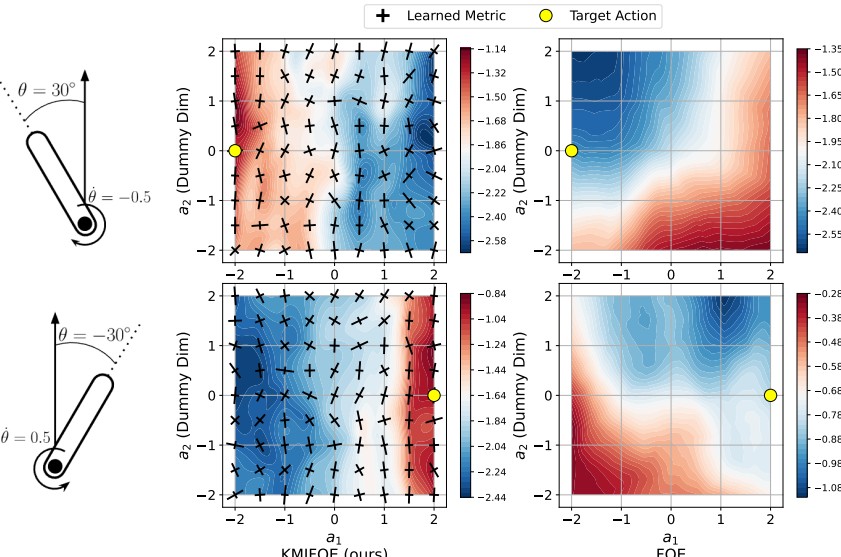

Figure 2: Visualization of $Q^\pi$ estimated by KMIFQE and FQE, along with the kernel metrics learned by KMIFQE in the modified Pendulum-v0 domain with one dummy action dimension. The original action dimension is $a_1$, and the dummy action dimension is $a_2$. The leftmost column illustrates the given states. The center column shows the Q-landscapes and metrics (black crosses) learned by KMIFQE. The rightmost column shows the Q-landscapes learned by FQE. The target actions at the given states are presented in yellow circles.

The performance of KMIFQE with learned metric and bandwidth, as well as baselines on the pendulum with one dummy action dimension are reported as root mean squared errors (RMSEs) in the first row of Table 1. KMIFQE outperforms the baselines that use OOD samples, and when metric learning is applied, the RMSEs are further reduced. The result supports the statement in Theorem 2 that the metric learning reduces the error bound.

Table 1: RMSEs of Baselines and KMIFQE with learned metric and bandwidth with one standard error on 10 seeds. Note that m-e=medium-expert, m-r=medium-replay.

| Dataset | known $\mu$ | KMIFQE | KMIFQE w/o Metric | SR-DICE | FQE |
|---|---|---|---|---|---|
| Pendulum with dummy dim | O | $\mathbf{0.115 \pm 0.018}$ | $0.250 \pm 0.032$ | $1.641 \pm 0.426$ | $8.126 \pm 2.962$ |
| Hopper-v2 | O | $\mathbf{0.023 \pm 0.006}$ | $0.034 \pm 0.009$ | $0.129 \pm 0.023$ | $0.083 \pm 0.011$ |
| HalfCheetah-v2 | O | $2.080 \pm 0.010$ | $2.549 \pm 0.017$ | $2.784 \pm 0.030$ | $\mathbf{1.637 \pm 0.051}$ |
| Walker2d-v2 | O | $\mathbf{0.032 \pm 0.008}$ | $0.048 \pm 0.009$ | $0.273 \pm 0.054$ | $241.319 \pm 49.248$ |
| Ant-v2 | O | $\mathbf{1.800 \pm 0.013}$ | $2.255 \pm 0.014$ | $1.996 \pm 0.030$ | $3.219 \pm 0.736$ |
| Humanoid-v2 | O | $\mathbf{0.246 \pm 0.010}$ | $0.293 \pm 0.021$ | $1.285 \pm 0.050$ | $8.860 \pm 8.196$ |
| hopper-m-e-v2 | X | $\mathbf{0.019 \pm 0.003}$ | $\mathbf{0.020 \pm 0.005}$ | $0.045 \pm 0.007$ | $0.033 \pm 0.010$ |
| halfcheetah-m-e-v2 | X | $0.418 \pm 0.016$ | $0.457 \pm 0.007$ | $0.239 \pm 0.025$ | $\mathbf{0.080 \pm 0.007}$ |
| walker2d-m-e-v2 | X | $\mathbf{0.036 \pm 0.006}$ | $\mathbf{0.038 \pm 0.006}$ | $0.115 \pm 0.017$ | $1.051 \pm 0.633$ |
| hopper-m-r-v2 | X | $\mathbf{0.536 \pm 0.099}$ | $\mathbf{0.517 \pm 0.120}$ | $0.849 \pm 0.052$ | $\mathbf{0.561 \pm 0.118}$ |
| halfcheetah-m-r-v2 | X | $\mathbf{4.698 \pm 0.044}$ | $4.765 \pm 0.026$ | $5.048 \pm 0.090$ | $6.394 \pm 1.769$ |
| walker2d-m-r-v2 | X | $\mathbf{1.364 \pm 0.052}$ | $\mathbf{1.360 \pm 0.025}$ | $1.523 \pm 0.061$ | $86.315 \pm 29.206$ |

## 5.2 CONTINUOUS CONTROL TASKS WITH A KNOWN BEHAVIOR POLICY

We compare the performance of KMIFQE with baselines on more complex continuous control tasks of MuJoCo (Todorov et al., 2012) with a known behavior policy. Deterministic target policies are trained by TD3 Fujimoto et al. (2018). For behavior policies, deterministic policies $\tilde{\mu}$ are trained with TD3 to achieve $70\% \sim 80\%$ performance (in undiscounted returns) of the target policy and are used for making the stochastic behavior policies $\mu(\mathbf{a} \mid \mathbf{s}) = N(\mathbf{a}; \tilde{\mu}(\mathbf{s}), (0.3a_{\max})^2 I)$. One million transitions are collected with the behavior policy as the dataset for each environment.

The performance of OPE methods on five MuJoCo environments reported in Table 1 shows that KMIFQE outperforms other baselines in all environments except HalfCheetah-v2 where FQE performs the best. Since HalfCheetah-v2 does not have an episode termination condition, the $Q^\pi$ that is being estimated may be relatively slowly changing w.r.t. changes in actions compared to the other environments, inducing low extrapolation error on the TD targets of FQE with OOD samples.

## 5.3 CONTINUOUS CONTROL TASKS WITH UNKNOWN MULTIPLE BEHAVIOR POLICIES

KMIFQE can be applied to a dataset sampled with unknown multiple behavior policies with a maximum likelihood estimated behavior policy. We evaluate KMIFQE on a D4RL (1) medium-expert dataset sampled from medium and expert level performance policies, and (2) medium-replay dataset collected from a single policy while it is being trained to achieve medium-level performance (Fu et al., 2020). The behavior policies used by KMIFQE are estimated as tanh-squashed mixture-of-Gaussians. For target policies, we use the means of stochastic expert policies provided by D4RL.

The last six rows of Table 1 show that KMIFQE outperforms the baselines except in halfcheetah-medium-expert-v2. The dataset may induce small extrapolation errors for the OPE methods that use OOD samples. First, half of the dataset is sampled with a stochastic expert policy, in which the mean value is the action selected by the target policy. Secondly, as discussed in Section 5.2, the $Q^\pi$ being estimated in the halfcheetah environment may vary slowly w.r.t. changes in actions, making the extrapolation error due to using OOD samples small for FQE and SR-DICE.

## 6 CONCLUSION

We presented KMIFQE, an off-policy evaluation (OPE) algorithm for deterministic reinforcement learning policies in continuous action spaces. KMIFQE learns the Q-function of a deterministic target policy for OPE in an in-sample manner through importance resampling (IR)-based Q-function update vector estimation, avoiding extrapolation error from OOD samples. IR is enabled by kernel-relaxing the deterministic target policy, and the estimation error from the relaxation is minimized by kernel metric learning. To learn the kernel metric, we first derived the MSE of an update vector estimation. Then, to we derived an optimal bandwidth that minimizes the MSE. Upon our observation that the bias is dominant in the MSE for a high-dimensional action space, the bandwidth-agnostic optimal metric matrix for bias reduction was derived as a closed-form solution. The optimal metric matrix is computed with the Hessian of the learned Q-function. Lastly, we analyzed the error bound of the learned Q-function by KMIFQE. Empirical studies show our KMIFQE outperforms baselines on the offline data sampled with known or unknown behavior policies.

## 7 ACKNOWLEDGEMENTS

This work was supported by IITP grant funded by MSIT (No.2020-0-00940, Foundations of Safe Reinforcement Learning and Its Applications to Natural Language Processing; No.2022-0-00311, Development of Goal-Oriented Reinforcement Learning Techniques for Contact-Rich Robotic Manipulation of Everyday Objects; No.2019-0-00075, AI Graduate School Program (KAIST)). Tri Wahyu Guntara was partially supported by the Hyundai Motor Chung Mong-Koo Foundation.

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

## A    DERIVATIONS AND PROOFS

### A.1    PROOF OF THEOREM 1

**Theorem 1.** *Under Assumption 1-2, the bias and variance of $\widehat{\Delta}_{IR}^K$ are:*

$$\text{Bias}[\widehat{\Delta}_{IR}^K] = h^2 \underbrace{\frac{\gamma}{2}\mathbb{E}_{\mathcal{D}\sim p_\mu}\left[\nabla_{\mathbf{a}'}^2 Q_{\bar{\theta}}(\mathbf{s}',\mathbf{a}')|_{\mathbf{a}'=\tilde{\pi}(\mathbf{s}')}\nabla_\theta Q_\theta(\mathbf{s},\mathbf{a})\right]}_{=:\mathbf{b}} + \mathcal{O}(h^4), \tag{19}$$

$$\text{Var}[\widehat{\Delta}_{IR}^K] = \frac{1}{nh^d}\underbrace{C(K)\mathbb{E}_{p_\mu}\left[\frac{(r+\gamma Q_{\bar{\theta}}(\mathbf{s}',\tilde{\pi}(\mathbf{s}'))-Q_\theta(\mathbf{s},\mathbf{a}))^2\|\nabla_\theta Q_\theta(\mathbf{s},\mathbf{a})\|_2^2}{\mu(\tilde{\pi}(\mathbf{s}')\mid\mathbf{s}')}\right]}_{=:v} + \mathcal{O}\left(\frac{1}{nh^{d-2}}\right), \tag{20}$$

*where $\text{Var}[\mathbf{z}] := \text{tr}[\text{Cov}(\mathbf{z},\mathbf{z})]$ for a vector $\mathbf{z}$, $\mathbf{b}$ is the bias constant vector, $v$ is the variance constant, $h^2\mathbf{b}$ is the leading-order bias, $\frac{v}{nh^d}$ is the leading-order variance, and $C(K) := \int K(\mathbf{z})^2 d\mathbf{z}$.*

*Proof.* With Assumptions 1-2,

**Derivation of the bias in Eq. (8):**

$$\text{Bias}[\hat{\Delta}_{IR}^K]$$

$$= \mathbb{E}_{\mathcal{D}\sim p_\mu}\left[\bar{w}^K\mathbb{E}_{\bar{x}\sim\rho^K}\left[\frac{1}{k}\sum_{i=1}^k\Delta(\bar{x}_i)\Big|\mathcal{D}\right]\right] - \mathbb{E}_{x\sim p_\pi}[\Delta(x)] \tag{21}$$

$$= \mathbb{E}_{\mathcal{D}\sim p_\mu}\left[\bar{w}^K\sum_{j=1}^n\frac{w^K(\mathbf{s}_j',\mathbf{a}_j')}{\sum_{i=1}^n w^K(\mathbf{s}_i',\mathbf{a}_i')}\Delta(x)\right] - \mathbb{E}_{x\sim p_\pi}[\Delta(x)] \tag{22}$$

$$= \mathbb{E}_{\mathcal{D}\sim p_\mu}\left[\frac{1}{n}\sum_{j=1}^n w^K(\mathbf{s}_j',\mathbf{a}_j')\Delta(x)\right] - \mathbb{E}_{x\sim p_\pi}[\Delta(x)] \tag{23}$$

$$= \mathbb{E}_{\mathcal{D}\sim p_\mu}\left[w^K(\mathbf{s}',\mathbf{a}')\Delta(x)\right] - \mathbb{E}_{x\sim p_\pi}[\Delta(x)] \tag{24}$$

$$= \mathbb{E}_{(\mathbf{s},\mathbf{a},r,\mathbf{s}')\sim p_\mu}\left[\int\frac{1}{h^d}K\left(\frac{\mathbf{a}'-\tilde{\pi}(\mathbf{s}')}{h}\right)(r+\gamma Q_{\bar{\theta}}(\mathbf{s}',\mathbf{a}')-Q_\theta(\mathbf{s},\mathbf{a}))\nabla_\theta Q_\theta(\mathbf{s},\mathbf{a})d\mathbf{a}'\right] \tag{25}$$

$$\quad - \mathbb{E}_{x\sim p_\pi}[\Delta(x)],$$

$$= \mathbb{E}_{(\mathbf{s},\mathbf{a},r,\mathbf{s}')\sim p_\mu}\left[\left(r+\gamma\left\{Q_{\bar{\theta}}(\mathbf{s}',\tilde{\pi}(\mathbf{s}'))+\frac{h^2}{2}\int K(\mathbf{z}')(\mathbf{z}')^\top \mathbf{H}_{\mathbf{a}'}Q_{\bar{\theta}}(\mathbf{s}',\mathbf{a}')|_{\mathbf{a}'=\tilde{\pi}(\mathbf{s}')}\mathbf{z}'d\mathbf{z}'\right\}\right.\right. \tag{26}$$

$$\quad \left.\left. - Q_\theta(\mathbf{s},\mathbf{a})\right)\nabla_\theta Q_\theta(\mathbf{s},\mathbf{a})\right] + \mathcal{O}(h^4) - \mathbb{E}_{x\sim p_\pi}[\Delta(x)]$$

$$= \mathbb{E}_{(\mathbf{s},\mathbf{a},r,\mathbf{s}')\sim p_\mu}\left[\left(r+\gamma\left\{Q_{\bar{\theta}}(\mathbf{s}',\tilde{\pi}(\mathbf{s}'))+\frac{h^2}{2}\text{tr}\left[\int K(\mathbf{z}')\mathbf{z}'(\mathbf{z}')^\top d\mathbf{z}'\,\mathbf{H}_{\mathbf{a}'}Q_{\bar{\theta}}(\mathbf{s}',\mathbf{a}')|_{\mathbf{a}'=\tilde{\pi}(\mathbf{s}')}\right]\right\}\right.\right. \tag{27}$$

$$\quad \left.\left. - Q_\theta(\mathbf{s},\mathbf{a})\right)\nabla_\theta Q_\theta(\mathbf{s},\mathbf{a})\right] + \mathcal{O}(h^4) - \mathbb{E}_{x\sim p_\pi}[\Delta(x)]$$

$$= \frac{\gamma h^2}{2}\mathbb{E}_{(\mathbf{s},\mathbf{a},r,\mathbf{s}')\sim p_\mu}\left[\nabla_{\mathbf{a}'}^2 Q_{\bar{\theta}}(\mathbf{s}',\mathbf{a}')|_{\mathbf{a}'=\tilde{\pi}(\mathbf{s}')}\nabla_\theta Q_\theta(\mathbf{s},\mathbf{a})\right] + \mathcal{O}(h^4), \tag{28}$$

where $\nabla_{\mathbf{a}'}^2$ and $\mathbf{H}_{\mathbf{a}'}$ are the Laplacian and Hessian operator w.r.t. the next action $\mathbf{a}'$ respectively, equality on Eq. (26) is obtained by applying Taylor expansion in Eq. (25), and change-of-variable $\mathbf{z}' = \frac{\mathbf{a}'-\tilde{\pi}(\mathbf{s}')}{h}$. The Taylor expansion is applied on $Q_{\bar{\theta}}(\mathbf{s}',\mathbf{a}')$ at $\mathbf{a}' = \tilde{\pi}(\mathbf{s}')$. The odd terms of the Taylor expansion are canceled out due to the symmetric property of the Gaussian kernel. For the equality in Eq. (28), we used the property of a Gaussian kernel $\int K(\mathbf{z}')\mathbf{z}'(\mathbf{z}')^\top d\mathbf{z}' = I$.

**Derivation of the variance in Eq. (9):**

$$\mathrm{Var}[\widehat{\Delta}_{IR}^K] = \underbrace{\mathbb{E}_{\mathcal{D}\sim p_\mu}\left[\mathrm{Var}_{\bar{x}\sim\rho^K}[\widehat{\Delta}_{IR}^K \mid \mathcal{D}]\right]}_{=:A} + \underbrace{\mathrm{Var}_{\mathcal{D}\sim p_\mu}\left[\mathbb{E}_{\bar{x}\sim\rho^K}[\widehat{\Delta}_{IR}^K \mid \mathcal{D}]\right]}_{=:B}, \tag{29}$$

$$A = \mathbb{E}_{\mathcal{D}\sim p_\mu}\left[(\bar{w}^K)^2 \,\mathrm{Var}_{\bar{x}\sim\rho^K}\left[\frac{1}{k}\sum_{j=1}^k \Delta(\bar{x}_j) \mid \mathcal{D}\right]\right] \tag{30}$$

$$= \mathbb{E}_{\mathcal{D}\sim p_\mu}\left[\frac{(\bar{w}^K)^2}{k}\,\mathrm{Var}_{\bar{x}\sim\rho^K}[\Delta(\bar{x})|\mathcal{D}]\right] \tag{31}$$

$$= \mathbb{E}_{\mathcal{D}\sim p_\mu}\left[\frac{(\bar{w}^K)^2}{k}\left\{\mathbb{E}_{\bar{x}\sim\rho^K}\left[\|\Delta(\bar{x})\|_2^2|\mathcal{D}\right] - \|\mathbb{E}_{\bar{x}\sim p_\mu}[\Delta(\bar{x})|\mathcal{D}]\|_2^2\right\}\right] \tag{32}$$

$$= \frac{1}{n^2 k}\mathbb{E}_{\mathcal{D}\sim p_\mu}\left[\left(\sum_{i=1}^n w^K(\mathbf{s}_i', \mathbf{a}_i')\right)^{\not\!\!2}\sum_{j=1}^n\left(\frac{w^K(\mathbf{s}_j', \mathbf{a}_j')}{\sum_{i=1}^n w^K(\mathbf{s}_i', \mathbf{a}_i')}\right)\|\Delta(x_j)\|_2^2 \right. \tag{33}$$

$$\left. - \left(\sum_{i=1}^n w^K(\mathbf{s}_i', \mathbf{a}_i')\right)^2\left\|\sum_{j=1}^n\frac{w^K(\mathbf{s}_j', \mathbf{a}_j')}{\sum_{i=1}^n w^K(\mathbf{s}_i', \mathbf{a}_i')}\Delta(x_j)\right\|_2^2\right]$$

$$= \frac{1}{n^2 k}\mathbb{E}_{\mathcal{D}\sim p_\mu}\left[\cancel{\sum_{j=1}^n (w^K(\mathbf{s}_j', \mathbf{a}_j'))^2\|\Delta(x_j)\|_2^2} + \sum_{i\neq j}^{n,n} w^K(\mathbf{s}_i', \mathbf{a}_i')w^K(\mathbf{s}_j', \mathbf{a}_j')\|\Delta(x_j)\|_2^2 \right. \tag{34}$$

$$\left. - \left(\cancel{\sum_{j=1}^n (w^K(\mathbf{s}_j', \mathbf{a}_j'))^2\|\Delta(x_j)\|_2^2} + \sum_{i\neq j}^{n,n} w^K(\mathbf{s}_i', \mathbf{a}_i')w^K(\mathbf{s}_j', \mathbf{a}_j')\Delta(x_i)^\top\Delta(x_j)\right)\right]$$

$$= \frac{n^2 - n}{n^2 k}\left\{\cancel{\mathbb{E}_{x\sim p_\mu}\left[w^K(\mathbf{s}', \mathbf{a}')\right]}\mathbb{E}_{x\sim p_\mu}\left[w^K(\mathbf{s}', \mathbf{a}')\|\Delta(x)\|_2^2\right] \right. \tag{35}$$

$$\left. - \mathbb{E}_{x\sim p_\mu}[w^K(\mathbf{s}', \mathbf{a}')\Delta(x)]^\top\mathbb{E}_{x\sim p_\mu}[w^K(\mathbf{s}', \mathbf{a}')\Delta(x)]\right\}$$

$$= \frac{n^2 - n}{n^2 k}\left\{\mathbb{E}_{x\sim p_\mu}\left[w^K(\mathbf{s}', \mathbf{a}')\|\Delta(x)\|_2^2\right] \right. \tag{36}$$

$$\left. - \mathbb{E}_{x\sim p_\mu}[w^K(\mathbf{s}', \mathbf{a}')\Delta(x)]^\top\mathbb{E}_{x\sim p_\mu}[w^K(\mathbf{s}', \mathbf{a}')\Delta(x)]\right\}$$

$$= \frac{n^2 - n}{n^2 k}\,\mathrm{Var}_{(\mathbf{s}, \mathbf{a}, r, \mathbf{s}')\sim p_\mu, \mathbf{a}'\sim\frac{1}{h^d}K\left(\frac{\mathbf{a}'-\bar{\pi}(\mathbf{s}')}{h}\right)}[\Delta(x)] \tag{37}$$

$$= \mathcal{O}\left(\frac{1}{k}\right), \tag{38}$$

$$B = \mathrm{Var}_{\mathcal{D}\sim p_\mu}[\mathbb{E}_{\bar{x}\sim\rho^K}[\widehat{\Delta}_{IR}^K|\mathcal{D}]] \tag{39}$$

$$= \mathrm{Var}_{\mathcal{D}\sim p_\mu}\left[\bar{w}^K\mathbb{E}_{\bar{x}\sim\rho^K}\left[\frac{1}{\cancel{k}}\cancel{\sum_{j=1}^k}\Delta(\bar{x}_j)\Big|\mathcal{D}\right]\right] \tag{40}$$

$$= \mathrm{Var}_{\mathcal{D}\sim p_\mu}\left[\frac{1}{n}\sum_{i=1}^n \cancel{w^K(\mathbf{s}_i', \mathbf{a}_i')}\left[\sum_{j=1}^n\frac{w^K(\mathbf{s}_j', \mathbf{a}_j')}{\sum_{i=1}^n\cancel{w^K(\mathbf{s}_i', \mathbf{a}_i')}}\Delta(x_j)\right]\right] \tag{41}$$

$$= \frac{1}{n}\,\mathrm{Var}_{x\sim p_\mu}\left[w^K(\mathbf{s}', \mathbf{a}')\Delta(x)\right] \tag{42}$$

$$= \frac{1}{n}\left\{\underbrace{\mathbb{E}_{x\sim p_\mu}[(w^K(\mathbf{s}', \mathbf{a}'))^2\|\Delta(x)\|_2^2]}_{=:X} - \underbrace{\|\mathbb{E}_{x\sim p_\mu}[w^K(\mathbf{s}', \mathbf{a}')\Delta(x)]\|_2^2}_{=\|\mathrm{Bias}[\widehat{\Delta}_{IR}]+\Delta_{TD}\|_2^2=\mathcal{O}(1)}\right\} \tag{43}$$

$$= \frac{X}{n} + \mathcal{O}\left(\frac{1}{n}\right), \tag{44}$$

$$X = \mathbb{E}_{(\mathbf{s},\mathbf{a},r,\mathbf{s}') \sim p_\mu} \left[ \int \frac{1}{h^{2d}} K \left( \frac{\mathbf{a}' - \tilde{\pi}(\mathbf{s}')}{h} \right)^2 \underbrace{\frac{(R(\mathbf{s},\mathbf{a}) + \gamma Q_{\bar{\theta}}(\mathbf{s}',\mathbf{a}') - Q_\theta(\mathbf{s},\mathbf{a}))^2}{\mu(\mathbf{a}'|\mathbf{s}')}}_{=:g(\mathbf{s},\mathbf{a},r,\mathbf{s}',\mathbf{a}')} \|\nabla_\theta Q_\theta(\mathbf{s},\mathbf{a})\|_2^2 d\mathbf{a}' \right]$$

(45)

$$= \mathbb{E}_{(\mathbf{s},\mathbf{a},r,\mathbf{s}') \sim p_\mu} \left[ \int \frac{K(\mathbf{z}')^2}{h^{2d}} g(\mathbf{s},\mathbf{a},r,\mathbf{s}',h\mathbf{z}' + \pi(\mathbf{s}')) \|\nabla_\theta Q_\theta(\mathbf{s},\mathbf{a})\|_2^2 h^d d\mathbf{z}' \right]$$

(46)

$$= \frac{C(K)}{h^d} \mathbb{E}_{(\mathbf{s},\mathbf{a},r,\mathbf{s}') \sim p_\mu} \left[ g(\mathbf{s},\mathbf{a},r,\mathbf{s}',\pi(\mathbf{s}')) \|\nabla_\theta Q_\theta(\mathbf{s},\mathbf{a})\|_2^2 \right] + \mathcal{O} \left( \frac{1}{h^{d-2}} \right),$$

(47)

where $C(K) := \int K(\mathbf{z})^2 d\mathbf{z}$, the equality in Eq. (46) is made by applying change-of-variable $\mathbf{z}' = \frac{\mathbf{a}' - \tilde{\pi}(\mathbf{s}')}{h}$ in Eq. (45), and the equality in Eq. (47) is made by applying Taylor expansion on $g(\mathbf{s},\mathbf{a},r,\mathbf{s}',\mathbf{a}')$ at $\mathbf{a}' = \tilde{\pi}(\mathbf{s}')$ in Eq. (46).

By plugging in Eq. (47) to Eq. (44),

$$B = \frac{C(K)}{nh^d} \mathbb{E}_{(\mathbf{s},\mathbf{a},r,\mathbf{s}') \sim p_\mu} \left[ \frac{(R(\mathbf{s},\mathbf{a}) + \gamma Q_{\bar{\theta}}(\mathbf{s}',\pi(\mathbf{s}')) - Q_\theta(\mathbf{s},\mathbf{a}))^2 \|\nabla_\theta Q_\theta(\mathbf{s},\mathbf{a})\|_2^2}{\mu(\tilde{\pi}(\mathbf{s}')|\,\mathbf{s}')} \right] + \mathcal{O} \left( \frac{1}{nh^{d-2}} \right).$$

(48)

By substituting Eq. (38) and Eq. (48) into Eq. (29), and since the mini-batch size $k$ is a constant multiple of $n$,

$$\therefore \mathrm{Var}[\widehat{\Delta}_{IR}^K] = \frac{C(K)}{nh^d} \mathbb{E}_{p_\mu} \left[ \frac{(R(\mathbf{s},\mathbf{a}) + \gamma Q_{\bar{\theta}}(\mathbf{s}',\tilde{\pi}(\mathbf{s}')) - Q_\theta(\mathbf{s},\mathbf{a}))^2 \|\nabla_\theta Q_\theta(\mathbf{s},\mathbf{a})\|_2^2}{\mu(\tilde{\pi}(\mathbf{s}')|\,\mathbf{s}')} \right] + \mathcal{O} \left( \frac{1}{nh^{d-2}} \right)$$

(49)

$\square$

## A.2 PROOF OF COROLLOARY 1

**Corollary 1.** *Under Assumption 1-2, MSE between the TD update vector estimate $\widehat{\Delta}_{IR}^K$ and the on-policy TD update vector $\Delta_{TD}$ is:*

$$\mathrm{MSE}(h,n,k,d) = \underbrace{h^4 \| \mathbf{b} \|_2^2 + \frac{v}{nh^d}}_{=:\mathrm{LOMSE}(h,n,d)} + \mathcal{O} \left( h^6 \right) + \mathcal{O} \left( \frac{1}{nh^{d-2}} \right),$$

(50)

*where* $\mathrm{LOMSE}(h,n,d)$ *is the leading-order MSE.*

*Proof.*

$$\mathrm{MSE}(h,n,k,d) = \left\| \mathrm{Bias} \left[ \widehat{\Delta}_{IR}^K \right] \right\|_2^2 + \mathrm{Var} \left[ \widehat{\Delta}_{IR}^K \right]$$

(51)

$$= \underbrace{h^4 \| \mathbf{b} \|_2^2 + \frac{v}{nh^d}}_{=:\mathrm{LOMSE}(h,n,d)} + \mathcal{O} \left( h^6 \right) + \mathcal{O} \left( \frac{1}{nh^{d-2}} \right).$$

(52)

$\square$

## A.3 PROOF OF PROPOSITION 1

**Proposition 1.** *The optimal bandwidth $h^*$ that minimizes the $LOMSE(h,n,d)$ is:*

$$h^* = \left( \frac{vd}{4n\| \mathbf{b} \|_2^2} \right)^{\frac{1}{d+4}}.$$

(53)

*Proof.* The LOMSE$(h, n, d)$ (Eq. (10)) minimizing optimal bandwidth $h^*$ is:

$$\frac{d}{dh}\left(\text{LOMSE}\left(h, n, d\right)\right) = 4h^3 \|\mathbf{b}\|_2^2 - \frac{vd}{nh^{d+1}}, \tag{54}$$

$$\therefore h^* = \left(\frac{vd}{4n\|\mathbf{b}\|_2^2}\right)^{\frac{1}{d+4}}. \tag{55}$$

$\square$

### A.4  PROOF OF PROPOSITION 2

**Proposition 2.** *Given optimal bandwidth $h^*$ (Eq. (11)), in high-dimensional action space, the squared L2-norm of leading-order bias $h^4\|\mathbf{b}\|_2^2$ dominates over the leading-order variance $\frac{v}{nh^d}$ in LOMSE (Eq. (10)). Furthermore,* LOMSE$(h^*, n, d)$ *approximates to* $\|\mathbf{b}\|_2^2$.

*Proof.* LOMSE$(h, n, d)$ in Eq. (10) is composed of leading-order bias LoBias$(h)$ and leading-order variance LoVar$(h, n, d)$:

$$\text{LoBias}(h) := h^2 \mathbf{b}, \tag{56}$$

$$\text{LoVar}(h, n, d) := \frac{v}{nh^d}. \tag{57}$$

The ratio of $\|\text{LoBias}(h^*)\|_2^2$ and LoVar$(h^*, n, d)$ with $h^*$ in Eq. (11) that together compose LOMSE$(h^*, n, d)$ as $d \to \infty$ is:

$$\lim_{d \to \infty} \frac{\|\text{LoBias}(h^*)\|_2^2}{\text{LoVar}(h^*, n, d)} = \lim_{d \to \infty} \frac{d}{4} = \infty. \tag{58}$$

From Eq. (58), it can be observed that the leading-order bias dominates over the leading-order variance in LOMSE when $d \gg 4$.

By plugging in $h^*$ (Eq. (11)) to the LOMSE in Eq. (10), and taking limit $d \to \infty$, we derive similar relation between LOMSE and the bias as in the work of Noh et al. (2017) that learned metric for kernel regression:

$$\text{LOMSE}(h^*, n, d) = n^{-\frac{4}{d+4}}\left(\left(\frac{d}{4}\right)^{\frac{4}{d+4}} + \left(\frac{4}{d}\right)^{\frac{d}{d+4}}\right)\|\mathbf{b}\|^{\frac{2d}{d+4}} v^{\frac{4}{d+4}}, \tag{59}$$

$$\therefore \lim_{d \to \infty} \text{LOMSE}(h^*, n, d) = \|\mathbf{b}\|_2^2. \tag{60}$$

From Eq. (59) and Eq. (60), it can be observed that for a high-dimensional action space with $d \gg 4$ and when the optimal bandwidth $h^*$ is used, the LOMSE can be approximated to $\|\mathbf{b}\|_2^2$.

$\square$

## A.5 Derivation of the bias with metric

Bias of the update vector estimation (Eq. (25)) with kernel inputs linearly transformed by $L(\mathbf{s}')$ $\left(\text{i.e., } \mathbf{z}' = \frac{L(\mathbf{s}')^\top (\mathbf{a}' - \tilde{\pi}(\mathbf{s}'))}{h}\right)$ is:

$\text{Bias}[\hat{\Delta}_{IR}^K]$

$$= \mathbb{E}_{(\mathbf{s},\mathbf{a},r,\mathbf{s}')\sim p_\mu} \left[\int \frac{1}{h^d} K\left(\frac{L(\mathbf{s}')^\top (\mathbf{a}' - \tilde{\pi}(\mathbf{s}'))}{h}\right) \left(r + \gamma Q_{\bar{\theta}}(\mathbf{s}', \mathbf{a}') - Q_\theta(\mathbf{s}, \mathbf{a})\right) \nabla_\theta Q_\theta(\mathbf{s}, \mathbf{a}) d\mathbf{a}'\right] \quad (61)$$

$$- \mathbb{E}_{x \sim p_\pi}[\Delta(x)],$$

$$= \mathbb{E}_{(\mathbf{s},\mathbf{a},r,\mathbf{s}')\sim p_\mu} \left[\left(r + \gamma \Big\{ Q_{\bar{\theta}}(\mathbf{s}', \tilde{\pi}(\mathbf{s}')) \right. \right. \quad (62)$$

$$+ \frac{h^2}{2} \int K(\mathbf{z}')(\mathbf{z}')^\top L(\mathbf{s}')^{-1} \mathbf{H}_{\mathbf{a}'} Q_{\bar{\theta}}(\mathbf{s}', \mathbf{a}')|_{\mathbf{a}'=\tilde{\pi}(\mathbf{s}')} L(\mathbf{s}')^{-\top} \mathbf{z}' d\mathbf{z}' \Big\}$$

$$\left. - Q_\theta(\mathbf{s}, \mathbf{a})\right) \nabla_\theta Q_\theta(\mathbf{s}, \mathbf{a})\right] + \mathcal{O}(h^4) - \mathbb{E}_{x \sim p_\pi}[\Delta(x)]$$

$$= \mathbb{E}_{(\mathbf{s},\mathbf{a},r,\mathbf{s}')\sim p_\mu} \left[\left(\frac{h^2 \gamma}{2} \text{tr}\left[L(\mathbf{s}')^{-\top} \underbrace{\int K(\mathbf{z}')\mathbf{z}'(\mathbf{z}')^\top d\mathbf{z}'}_{=I} L(\mathbf{s}')^{-1} \mathbf{H}_{\mathbf{a}'} Q_{\bar{\theta}}(\mathbf{s}', \mathbf{a}')|_{\mathbf{a}'=\tilde{\pi}(\mathbf{s}')}\right]\right. \quad (63)$$

$$\left.\Big) \nabla_\theta Q_\theta(\mathbf{s}, \mathbf{a})\right] + \mathcal{O}(h^4)$$

$$= h^2 \cdot \underbrace{\frac{\gamma}{2} \mathbb{E}_{(\mathbf{s},\mathbf{a},r,\mathbf{s}')\sim p_\mu} \left[\text{tr}\left[A(\mathbf{s}')^{-1} \mathbf{H}_{\mathbf{a}'} Q_{\bar{\theta}}(\mathbf{s}', \mathbf{a}')|_{\mathbf{a}'=\tilde{\pi}(\mathbf{s}')}\right] \nabla_\theta Q(\mathbf{s}, \mathbf{a})\right]}_{=:\mathbf{b}_A}. \quad (64)$$

For the second equality in Eq. (62), we used Taylor expansion on $Q_{\bar{\theta}}(\mathbf{s}', \mathbf{a}')$ at $\mathbf{a}' = \tilde{\pi}(\mathbf{s}')$. The odd terms of the Taylor expansion are canceled out due to the symmetric property of the Gaussian kernel. For the last equality, we use the relation $A(\mathbf{s}') = L(\mathbf{s}')L(\mathbf{s}')^\top$.

$$\therefore \|\mathbf{b}_A\|_2^2 = \frac{\gamma^2}{4} \|\mathbb{E}_{\mathcal{D}\sim p_\mu} \left[\text{tr}\left[A(\mathbf{s}')^{-1} \mathbf{H}_{\mathbf{a}'} Q_{\bar{\theta}}(\mathbf{s}', \mathbf{a}')|_{\mathbf{a}'=\tilde{\pi}(\mathbf{s}')}\right] \nabla_\theta Q(\mathbf{s}, \mathbf{a})\right]\|_2^2.$$

## A.6 PROOF OF THEOREM 2

**Theorem 2.** *Under Assumption 3, denoting $m$ iterative applications of $T$ and $T_K$ to an arbitrary $Q$ as $T^m Q$ and $T_K^m Q$ respectively, and $h$ and $A(\mathbf{s}')$ as an arbitrary bandwidth and metric respectively, the following holds:*

$$\|Q^\pi - \lim_{m \to \infty} T_K^m Q\|_\infty \leq \frac{\gamma \xi}{1 - \gamma}, \tag{65}$$

$$\xi := \max_{m, \{\mathbf{s}, \mathbf{a}\} \in S \times A} \frac{h^2}{2} \mathbb{E}_{\mathbf{s}' \sim P(\mathbf{s}' \mid \mathbf{s}, \mathbf{a})} \left[ \left| \mathrm{tr}\left( A(\mathbf{s}')^{-1} \, \mathbf{H}_{\mathbf{a}'} \, T_K^m Q(\mathbf{s}', \mathbf{a}')|_{\mathbf{a}' = \tilde{\pi}(\mathbf{s}')} \right) \right| \right] + |\mathcal{O}((h)^4)|. \tag{66}$$

*Proof.* Derive $\|TQ - T_K Q\|_\infty / \gamma$ w.r.t. $h$ and $A(\mathbf{s}')$ by using Taylor expansion at $\mathbf{a}' = \tilde{\pi}(\mathbf{s}')$:

$$\|TQ - T_K Q\|_\infty / \gamma$$

$$= \max_{\{\mathbf{s}, \mathbf{a}\} \in S \times A} \left| \mathbb{E}_{\mathbf{s}' \sim P(\mathbf{s}' \mid \mathbf{s}, \mathbf{a})} \left[ \int K\left( \frac{L_Q^*(\mathbf{s}')^\top (\mathbf{a}' - \tilde{\pi}(\mathbf{s}'))}{h_Q^*} \right) Q(\mathbf{s}', \mathbf{a}') d\mathbf{a}' - Q(\mathbf{s}', \tilde{\pi}(\mathbf{s}')) \right] \right|$$

$$= \max_{\{\mathbf{s}, \mathbf{a}\} \in S \times A} \left| \mathbb{E}_{\mathbf{s}' \sim P(\mathbf{s}' \mid \mathbf{s}, \mathbf{a})} \left[ \int K(\mathbf{z}') \left( \frac{(h_Q^*)^2}{2} (\mathbf{z}')^\top L_Q^*(\mathbf{s}')^{-1} H_{\mathbf{a}'} Q(\mathbf{s}', \mathbf{a}')|_{\mathbf{a}' = \tilde{\pi}(\mathbf{s}')} L_Q^*(\mathbf{s}')^{-\top} \mathbf{z}' \right. \right. \right.$$

$$\left. \left. \left. + \mathcal{O}((h_Q^*)^4) \right) d\mathbf{z}' \right] \right|$$

$$= \max_{\{\mathbf{s}, \mathbf{a}\} \in S \times A} \left| \mathbb{E}_{\mathbf{s}' \sim P(\mathbf{s}' \mid \mathbf{s}, \mathbf{a})} \left[ \frac{(h_Q^*)^2}{2} \mathrm{tr}\left( A_Q^*(\mathbf{s}')^{-1} H_{\mathbf{a}'} Q(\mathbf{s}', \mathbf{a}')|_{\mathbf{a}' = \tilde{\pi}(\mathbf{s}')} \right) + \mathcal{O}((h_Q^*)^4) \right] \right|$$

$$\leq \underbrace{\max_{\{\mathbf{s}, \mathbf{a}\} \in S \times A} \frac{(h_Q^*)^2}{2} \mathbb{E}_{\mathbf{s}' \sim P(\mathbf{s}' \mid \mathbf{s}, \mathbf{a})} \left[ \left| \mathrm{tr}\left( A_Q^*(\mathbf{s}')^{-1} H_{\mathbf{a}'} Q(\mathbf{s}', \mathbf{a}')|_{\mathbf{a}' = \tilde{\pi}(\mathbf{s}')} \right) \right| \right] + |\mathcal{O}((h_Q^*)^4)|}_{=: \xi} \tag{67}$$

where we used $\mathbf{z}' := \frac{L(\mathbf{s}')^\top (\mathbf{a}' - \tilde{\pi}(\mathbf{s}'))}{h}$, symmetricity of kernel $K(\mathbf{z}') = K(-\mathbf{z}')$, and $\int K(\mathbf{z}') \mathbf{z}' \mathbf{z}'^\top d\mathbf{z}' = I$.

Next, we conjecture that $\|T^m Q - T_K^m Q\|_\infty \leq \sum_{i=1}^m \gamma^i \xi$,

(i) For $m = 1$, the conjecture is true by the definition of $\xi$ in Eq. (67).

(ii) Assuming that the conjecture is true for an arbitrary $m = j$,

$$\|T^j Q - T_K^j Q\|_\infty \leq \sum_{i=1}^j \gamma^i \xi, \tag{68}$$

$$\|T^{j+1} Q - T_K^{j+1} Q\|_\infty \leq \|(T - T_K) T_K^j Q\|_\infty + \|T(T^j Q - T_K^j Q)\|_\infty \tag{69}$$

$$\leq \gamma \xi + \sum_{i=1}^j \gamma^{i+1} \xi \tag{70}$$

$$= \sum_{i=1}^{j+1} \gamma^i \xi \tag{71}$$

$$\tag{72}$$

$\therefore$ From (i) and (ii), the conjecture is proved to be true. Using the relation $T^\infty Q = Q^\pi$, Eq.(5) can be derived. Eq.(6) can be derived similarly.

$\square$

# B PSEUDO CODE FOR KMIFQE

---

**Algorithm 1** Kernel Metric Learning for In-Sample Fitted Q Evaluation

---

**Input:** Offline data $\mathcal{D} = \{\mathbf{s}_i, \mathbf{a}_i, r_i, \mathbf{s}'_i, \mathbf{a}'_i\}_{i=1}^n$, stochastic behavior policy $\mu$, deterministic target policy $\tilde{\pi}$, Q-function $Q_\theta$ with parameter $\theta$, target network $Q_{\bar{\theta}}$ parameterized by $\bar{\theta}$, Gaussian kernel $K$, initial linear transformation matrix $L(\mathbf{s}_i) = I$ for all $i$, learning rate $\alpha$, number of TD update iterations $N$, mini-batch size $k$.

**Output:** Estimated target policy value $\widehat{V}(\pi)$.

1: **for** until convergence **do**
2:    **for** $N$ steps **do**
3:       Update the bandwidth $h = \left( \frac{vd}{4n\|\mathbf{b}\|_2^2} \right)^{\frac{1}{d+4}}$ (Eq. (11)),

$$\text{where } \mathbf{b} := \frac{\gamma}{2k} \sum_{i=1}^k \nabla_{\mathbf{a}'}^2 Q_{\bar{\theta}}(\mathbf{s}'_i, \mathbf{a}')|_{\mathbf{a}'=\tilde{\pi}(\mathbf{s}'_i)} \nabla_\theta Q_\theta(\mathbf{s}_i, \mathbf{a}_i),$$

$$v := \frac{C(K)}{k} \sum_{i=1}^k \left[ \frac{(r_i + \gamma Q_{\bar{\theta}}(\mathbf{s}'_i, \tilde{\pi}(\mathbf{s}'_i)) - Q_\theta(\mathbf{s}_i, \mathbf{a}_i))^2 \|\nabla_\theta Q_\theta(\mathbf{s}_i, \mathbf{a}_i)\|_2^2}{\mu(\tilde{\pi}(\mathbf{s}'_i) \mid \mathbf{s}'_i)} \right],$$

$$C(K) := \int K(\mathbf{z})^2 d\mathbf{z}.$$

4:       Compute IS ratio for each sample $i$: $w^K(\mathbf{s}'_i, \mathbf{a}'_i) := \frac{1}{h^d \mu(\mathbf{a}'_i \mid \mathbf{s}'_i)} K\left( \frac{L(\mathbf{s}'_i)^\top (\mathbf{a}'_i - \tilde{\pi}(\mathbf{s}'_i))}{h} \right)$
5:       Compute resampling probability for each sample: $\rho_i^K = \frac{w^K(\mathbf{s}'_i, \mathbf{a}'_i)}{\sum_{j=1}^n w^K(\mathbf{s}'_j, \mathbf{a}'_j)}$
6:       Sample $k$ transitions with $\rho^K$: $\bar{x}_j \overset{\rho^K}{\sim} \{x_1, \ldots, x_n\}$, where $x_i := \{\mathbf{s}_i, \mathbf{a}_i, r_i, \mathbf{s}'_i, \mathbf{a}'_i\}$
7:       Compute the estimated expectation of the update vector $\widehat{\Delta}_{IR}^K = \frac{\bar{w}^K}{k} \sum_{j=1}^k \Delta(\bar{x}_j)$ in Eq. (7)
8:       Update $\theta$

$$\theta \leftarrow \theta + \alpha \widehat{\Delta}_{IR}^K$$

9:    **end for**
10:   Update $\bar{\theta}$

$$\bar{\theta} \leftarrow \theta$$

11:   Update linear transformation matrix $L(\mathbf{s}'_i)$ $(A(\mathbf{s}'_i) = L(\mathbf{s}'_i)L(\mathbf{s}'_i)^\top)$ for each sample

$$L(\mathbf{s}_i) = \alpha(\mathbf{s}'_i)^{\frac{1}{2}} [U_+(\mathbf{s}'_i)U_-(\mathbf{s}'_i)] \begin{pmatrix} d_+(\mathbf{s}'_i)\Lambda_+(\mathbf{s}'_i) & 0 \\ 0 & -d_-(\mathbf{s}'_i)\Lambda_-(\mathbf{s}'_i) \end{pmatrix}^{\frac{1}{2}},$$

$$\text{where } \alpha(\mathbf{s}'_i) := \left| \begin{pmatrix} d_+(\mathbf{s}'_i)\Lambda_+(\mathbf{s}'_i) & 0 \\ 0 & -d_-(\mathbf{s}'_i)\Lambda_-(\mathbf{s}'_i) \end{pmatrix} \right|^{-1/(d_+(\mathbf{s}'_i)+d_-(\mathbf{s}'_i))}.$$

12: **end for**
13: Estimate $\widehat{V}(\pi) = (1-\gamma)\frac{1}{m} \sum_{j=1}^m Q_\theta(\mathbf{s}_{0,j}, \tilde{\pi}(\mathbf{s}_{0,j}))$,

$$\text{where } \{\mathbf{s}_{0,j}\}_{j=1}^m \text{ is the set of initial states in } \mathcal{D}.$$

---

## C  EXPERIMENT DETAILS

### C.1  PENDULUM WITH DUMMY ACTION DIMENSIONS

**Environment**  The Pendulum-v0 environment in OpenAI Gym (Brockman et al., 2016) is modified to have d-dimensional actions. Among the d-dimensional actions, only the first dimension, which is the original action dimension in the Pendulum-v0 environment, is used for computing the next state transitions and rewards. The other (d - 1) action dimensions are unrelated to next state transitions and rewards. All of the dimensions have an action range equal to that of the original action space, which is $[-a_{\max}, a_{\max}]$, where $a_{\max} = 2$. The episode length is 200 steps. The discount factor of 0.95 is used.

**Target policy**  We train a TD3 policy $\tilde{\pi}_1$ to be near-optimal on the original Pendulum-v0 environment which has only one action dimension and use it to make a target policy. The target policy $\tilde{\pi}$ is made with $\tilde{\pi}_1(\mathbf{s})$:

$$\tilde{\pi}(\mathbf{s}) = \begin{bmatrix} \tilde{\pi}_1(\mathbf{s}) \\ 0 \\ \vdots \\ 0 \end{bmatrix}.$$

The policy value of $\pi_0$ estimated from the rollout of 1500 episodes is $-3.791$. Since the dummy action dimensions do not have an effect on next state transitions and rewards, the policy values of $\tilde{\pi}_1$ and $\tilde{\pi}$ are the same.

**Behavior policy**  For the behavior policy, a TD3 policy $\tilde{\mu}_1$ is deliberately trained to be inferior to $\tilde{\pi}_1$ and is used to make a stochastic behavior policy. The behavior policy $\mu$ made with $\tilde{\mu}_1$ is as follows:

$$\mu(\mathbf{a} \,|\, \mathbf{s}) = \left\{ 0.8 N(a_1 | \tilde{\mu}_1(\mathbf{s}), (0.5 a_{\max})^2) + 0.2 U(a_1 | -a_{\max}, a_{\max}) \right\} \prod_{i=2}^{d} U(a_i | -a_{\max}, a_{\max}), \quad (73)$$

where a uniform density with a range of $[-a_{\max}, a_{\max}]$ for $i^{\text{th}}$ action dimension is denoted as $U(a_i | -a_{\max}, a_{\max})$. For the first action dimension, the mixture of 80% Gaussian density and 20% uniform density is used. For the other dimensions, actions are sampled from the uniform densities. The behavior policy is similar to the one used in the work of Fujimoto et al. (2021).

The policy values of $\mu$ and $\tilde{\mu}_1$ are estimated from the rollout of 1500 episodes and reported in Table 3. Since the dummy action dimensions are unrelated to next state transitions and rewards, the behavior policy value does not change as the number of dummy action dimensions changes.

Table 2: Policy values

|  | Modified Pendulum-v0 |
|---|---|
| $R(\tilde{\mu}_1)$ | -453.392 |
| $R(\pi)$ | -142.553 |
| $V(\tilde{\mu}_1)$ | -4.413 |
| $V(\mu)$ | -5.080 |
| $V(\pi)$ | -3.971 |

**Dataset**  Half a million transitions are sampled with the behavior policy $\mu$ for all experiments using the modified pendulum environment with dummy action dimensions.

**Target policy value evaluation with OPE methods**  For KMIFQE and FQE, target policy values are estimated as $\widehat{V}(\pi) = (1 - \gamma) \frac{1}{m} \sum_{i=1}^{m} Q_\theta(\mathbf{s}_{0,i}, \tilde{\pi}(\mathbf{s}_{0,i}))$, where $m$ is the number of episodes in the dataset. For SR-DICE, the target policy values are estimated with 10k transitions randomly sampled from the data.

**Network architecture**   Our algorithm and FQE use the same architecture of a network of 2 hidden layers with 256 hidden units. For SR-DICE, we use the network architecture used in the work of Fujimoto et al. (2021). The encoder network of SR-DICE uses one hidden layer with 256 hidden units which outputs a feature vector of 256. The decoder network of SR-DICE uses one hidden layer with 256 hidden units for both next state and action decoders and uses a linear function of the feature vector for the reward decoder. The successor representation network of SR-DICE is composed of 2 hidden layers with 256 hidden units, and SR-DICE also uses 256 hidden units for the density ratio weights. We use the SR-DICE and FQE implementations in `https://github.com/sfujim/SR-DICE`.

**Hyperparameters**   Following the hyperparameter settings used in the work (Fujimoto et al., 2021), all networks are trained with Adam optimizer (Kingma & Ba, 2014) with the learning rate of $3e - 4$. For mini-batch sizes, the encoder-decoder network, and successor representation network of SR-DICE, as well as FQE, use a mini-batch size of 256. For the learning of density ratio in SR-DICE and our algorithm, we use a mini-batch size of 1024. FQE and SR-DICE use update rate $\tau = 0.005$ for the soft update of the target critic network and target successor representation network respectively. For our proposed method, target critic network is hard updated every 1000 iterations. We clip the behavior policy density value at the target action by 1e-5 when the density value is below the clipping value for numerical stability when evaluating $v$ defined in Eq. (9). The IS ratios are clipped to be in the range of $[0.001, 2]$ to lower the variance of its estimations (Kallus & Zhou, 2018; Munos et al., 2016; Han & Sung, 2019). The clipping range is selected by grid search on Hopper-v2 domain from the cartesian product of minimum IS ratio clip values {1e-5, 1e-3, 1e-1} and maximum IS ratio clip values {1, 2, 10}.

**Computational resources used and KMIFQE train time**   One i7 CPU with one NVIDIA Titan Xp GPU runs KMIFQE for two million train steps in 5 hours.

## C.2   Continuous control tasks with a known behavior policy

**Environment**   MuJoCo environments (Todorov et al., 2012) of Hopper-v2 (d=3), HalfCheetah-v2 (d=6), and Walker2D-v2 (d=6), Ant-v2 (d=8), and Humanoid-v2 (d=17) are used. The maximum episode length of the environments is 1000 steps. HalfCheetah-v2 does not have a termination condition that terminates an episode before it reaches 1000 steps. But the other two environments may terminate before reaching the maximum episode length when the agent falls. The discount factor of 0.99 is used. The action range for all environments and for all action dimensions is in $[-a_{\max}, a_{\max}]$, where $a_{\max} = 1$ except Humanoid-v2 which is $a_{\max} = 0.4$.

**Target policy**   We train a TD3 policy $\pi$ to be near-optimal and use it as a target policy. The target policy values estimated with the rollout of 1000 episodes are presented in Table 3.

**Behavior policy**   For the behavior policy of each environment of HalfCheetah-v2, Hopper-v2, and Walker2D-v2, a TD3 policies $\tilde{\mu}$ are deliberately trained to be inferior to a target policy, to achieve about 70%~80% of the undiscounted return $(R)$ of $\pi$. Then, $\tilde{\mu}$ is used to make the Gaussian behavior policy $\mu(\mathbf{a} \,|\, \mathbf{s}) = N(\mathbf{a} \,|\, \tilde{\mu}(\mathbf{s}), (0.3 a_{\max})^2 I)$. The policy values of the behavior policy $\mu$ and $\tilde{\mu}$ for each environment are estimated with the rollout of 1000 episodes, and the policy values are reported in Table 3.

**Target policy value evaluation with OPE methods**   For KMIFQE and FQE, target policy values are estimated as $\widehat{V}(\pi) = (1 - \gamma) \frac{1}{m} \sum_{i=1}^{m} Q_\theta(\mathbf{s}_{0,i}, \tilde{\pi}(\mathbf{s}_{0,i}))$, where $m$ is the number of episodes in the dataset. For SR-DICE, the target policy values are estimated with 10k transitions randomly sampled from the data.

**Dataset**   One million transitions are sampled with the behavior policy $\mu$ for all experiments.

**Target policy value evaluation with OPE methods**   For KMIFQE and FQE, target policy values are estimated as $\widehat{V}(\pi) = (1 - \gamma) \frac{1}{m} \sum_{i=1}^{m} Q(\mathbf{s}_{0,i}, \tilde{\pi}(\mathbf{s}_{0,i}))$, where $m$ is the number of episodes in the dataset. For SR-DICE, the target policy values are estimated with 10k transitions randomly sampled from the data.

Table 3: Policy values

|  | HalfCheetah-v2 | Hopper-v2 | Walker2D-v2 | Ant-v2 | Humnaoid-v2 |
|---|---|---|---|---|---|
| $R(\tilde{\mu})$ | 7591.245 | 2054.226 | 3032.198 | 3736.888 | 3912.369 |
| $R(\pi)$ | 10495.003 | 2600.161 | 4172.839 | 5034.756 | 5303.486 |
| $V(\tilde{\mu})$ | 5.716 | 2.553 | 2.595 | 3.439 | 4.848 |
| $V(\mu)$ | 4.211 | 2.506 | 2.605 | 1.886 | 4.396 |
| $V(\pi)$ | 7.267 | 2.571 | 2.693 | 4.396 | 5.211 |

**Network architecture** The network architectures used for the MuJoCo experiment are the same as that of the architectures used in the experiments on pendulum environment with dummy action dimensions.

**Hyperparameters** The hyperparameters used for the MuJoCo experiments are the same as that of the experiments on the pendulum environment with dummy action dimensions except the mini-batch size used for the learning of density ratio in SR-DICE and our algorithm is 2048 as in the work of Fujimoto et al. (2021). For our proposed method, the IS ratios are dimension-wise clipped in the range of $[0.001, 2]$ to lower the variance of its estimations (Han & Sung, 2019).

**Computational resources used and KMIFQE train time** One i7 CPU with one NVIDIA Titan Xp GPU runs KMIFQE for one million train steps in 5 hours.

## C.3 CONTINUOUS CONTROL TASKS WITH UNKNOWN MULTIPLE BEHAVIOR POLICIES

**Dataset** Among the D4RL dataset (Fu et al., 2020) we used halfcheetah-medium-expert-v2, hopper-medium-expert-v2, walker2d-medium-expert-v2, halfcheetah-medium-replay-v2, hopper-medium-replay-v2, walker2d-medium-replay-v2. The discount factor of 0.99 is used. The action range for all environments and for all action dimensions is in $[-a_{max}, a_{max}]$, where $a_{max} = 1$.

**Target policy** We use the mean values of the stochastic expert policies provided by D4RL dataset (Fu et al., 2020) as deterministic target policies.

**Target policy value evaluation with OPE methods** For KMIFQE and FQE, target policy values are estimated as $\widehat{V}(\pi) = (1 - \gamma)\frac{1}{m}\sum_{i=1}^{m} Q_\theta(\mathbf{s}_{0,i}, \tilde{\pi}(\mathbf{s}_{0,i}))$, where $m$ is the number of episodes in the dataset. For SR-DICE, the target policy values are estimated with 10k transitions randomly sampled from the data.

**Maximum likelihood estimation of behavior policies for KMIFQE** Behavior policies are maximum likelihood estimated tanh-squashed mixture-of-Gaussian (MoG) models. The MoGs are squashed by tanh to restrict the support of the behavior policy to the action range of $[-a_{max}, a_{max}]$ for each action dimension. We fit tanh-squashed MoGs with mixture number of 10, 20, 30, 40 and select a mixture number by cross-validation for each dataset. The validation set is the 10% of the data, and rest of the data is used for training. The validation log-likelihood for each mixture number is reported in Table 4 with one standard error.

**Network architecture** The network architectures used for the D4RL experiment are the same as that of the architectures used in the experiments on pendulum environment with dummy action dimensions.

**Hyperparameters** The hyperparameters used for the MuJoCo experiments are the same as those of the experiments on the MuJoCo domain (Todorov et al., 2012) with a known behavior policy except that since the dimension-wise clipping is not possible, we clip the overall IS ratios. Furthermore, we clip the behavior policy density value at the target action by 1e-3 when the density value is below the clipping value for numerical stability when evaluating $v$.

Table 4: Validation log-likelihood of estimated behavior policies.

| Dataset | Number of Mixtures | | | |
|---|---|---|---|---|
| | 10 | 20 | 30 | 40 |
| hopper-m-e | $2.089 \pm 0.002$ | $2.114 \pm 0.002$ | $2.123 \pm 0.003$ | $\mathbf{2.125 \pm 0.002}$ |
| halfcheetah-m-e | $6.581 \pm 0.003$ | $\mathbf{6.591 \pm 0.003}$ | $6.583 \pm 0.003$ | $6.578 \pm 0.003$ |
| walker2d-m-e | $5.610 \pm 0.002$ | $5.626 \pm 0.005$ | $\mathbf{5.630 \pm 0.005}$ | $5.629 \pm 0.005$ |
| hopper-m-r | $0.767 \pm 0.006$ | $0.787 \pm 0.007$ | $0.797 \pm 0.006$ | $\mathbf{0.800 \pm 0.005}$ |
| halfcheetah-m-r | $\mathbf{5.315 \pm 0.012}$ | $5.303 \pm 0.012$ | $5.281 \pm 0.016$ | $5.271 \pm 0.014$ |
| walker2d-m-r | $\mathbf{2.508 \pm 0.007}$ | $2.492 \pm 0.013$ | $2.480 \pm 0.016$ | $2.472 \pm 0.011$ |

**Computational resources used and KMIFQE train time**    One i7 CPU with one NVIDIA Titan Xp GPU runs KMIFQE for one million train steps in 5 hours.

