# OpenReview forum: "Kernel Metric Learning for In-Sample Off-Policy Evaluation of Deterministic RL Policies"
_ICLR.cc/2024/Conference — ICLR 2024 spotlight_

### Official Review · Reviewer_pubB · 2023-10-29

**Soundness:** 3 good
**Presentation:** 3 good
**Contribution:** 2 fair
**Rating:** 6
**Confidence:** 3

**Summary:**

The paper extends in-sample OPE methods to deterministic target policies by using the kernel approximation. The paper calculates the bias and variance of the estimation error resulting from this relaxation and present analytical solutions for the ideal kernel metric. Through empirical study, it demonstrate superior performance.

**Strengths:**

The paper is easy to follow, solid, and studies an important problem

**Weaknesses:**

The novelty is marginal, as the key components (such as kernel relaxation, in-sample learning, and metric learning) are standard. The main contribution is on the theoretical derivation.

**Questions:**

It would help to comment on how to extend the method to policy learning and discuss recent words e.g. "Singularity-aware Reinforcement Learning" and "Policy learning "without'' overlap: Pessimism and generalized empirical Bernstein's inequality".

Besides, as your key contribution, it might help to highlight how you do metric learning (currently it takes efforts to find).

---

> ### Author Response · Authors · 2023-11-15
> **Response to Reviewer $\text{\textcolor{red}{pubB}}$**
>
> Thank you for taking the time to provide thoughtful feedback on our work. Please let us know if any of our responses below need further clarification.
>
> $ $
>
> **W1. Novelty of our work**
>
> Performing in-sample off-policy evaluation (OPE) of a deterministic policy is challenging because in-sample OPE necessitates importance sampling (IS), while the IS ratios are zeros for a deterministic target policy. To address this issue, we utilized kernel relaxation and metric learning, which have not been used for OPE  in MDP settings. Our novelty lies in the integration of kernel relaxation, metric learning, and in-sample OPE to address the issue. To integrate the components, we derived the MSE of the in-sample estimated Q update vector regarding kernel relaxation and metric learning in MDP settings. To the best of our knowledge, our method is the first in-sample OPE algorithm for evaluating deterministic target policies. We believe our work’s novelty is also in creating the theoretical guarantee of the proposed method.
>
>
> $ $
>
> **Q1. How to extend our method to policy learning**
>
> Extending our method to policy improvement (PI) is our future work. Naive application of our work on PI will result in an actor-critic algorithm for offline RL where we evaluate the target policy with the critic learned by our method and improve policy by some PI methods such as the one used by TD3+BC [Fujimoto et al., 2021].
>
> The reviewer referred to the papers that have not been published yet. However, in considering how to extend our work to PI regarding those works, work by Jin et al. [Jin et al., 2023] is an offline contextual bandit policy learning algorithm that works on discrete action spaces and tries to learn the policy that has the highest Q-estimation while avoiding learning policy that has high uncertainty (or variance) in the Q-estimation. The variance comes from a large IS ratio due to small behavior policy density values. Because our method also uses IS, we may adopt their method when we expand our work to offline RL. The work by Chen et al. [Chen et al., 2023] is an offline policy iteration algorithm that works on infinite horizon MDPs with continuous states and actions. Upon analysis of an OPE error, they decomposed the error into two parts using Lebesgue’s decomposition theorem: 1) absolutely continuous part w.r.t. data distribution where importance sampling can be applied, 2) singular part (e.g., Dirac measure, deterministic target policy). For the singular part, they used maximum mean discrepancy to make the upper bound of the OPE error. Since our method is developed for OPE of deterministic policies, our method may be integrated into the singular part and be used for PI.
>
> $ $
>
> [Fujimoto et al., 2021] Fujimoto et al. “A minimalist approach to offline reinforcement learning” NeurIPS 2021.
>
> [Jin et al., 2023] Jin et al. “Policy learning “without” overlap: Pessimism and generalized empirical Bernstein’s inequality” Arxiv 2023.
>
> [Chen et al., 2023] Chen et al. “STEEL: Singularity-aware reinforcement learning”. ArXiv 2023.
>
>
>
> $ $
>
> **Q2. The detailed procedure on metric learning**
>
> To clear up how metric learning is done in our method, we explicitly state in the last paragraph of section 4.2 that the pseudo-code of our algorithm in Algorithm 1 in Appendix B, and modified Algorithm 1 to contain more details on metric learning.

---

> > ### Comment · Reviewer_pubB · 2023-11-21
> >
> > Thanks for the response. I find it helpful. I raised my score, conditioned on the discussions on novelty and policy optimization being incorporated to the manuscript.

---

> > > ### Author Response · Authors · 2023-11-22
> > > **Thank you for the constructive feedback**
> > >
> > > Thank you for the constructive feedback. We are glad to hear that you found our response helpful. We added an explanation of our novelty in the last paragraph of the Introduction section and the paragraph of the Conclusion section. In the Conclusion section, we also mentioned how to use our method in policy optimization. If you have further concerns, please let us know.

---

### Official Review · Reviewer_bgFP · 2023-10-31

**Soundness:** 3 good
**Presentation:** 3 good
**Contribution:** 3 good
**Rating:** 8
**Confidence:** 3

**Summary:**

The paper proposes Kernel Metric learning for In-sample Fitted Q Evaluation (KMIFQE), for off-policy evaluation of deterministic target policies in in-sample learning in continuous control tasks. The issue with the evaluation of a deterministic policy is that the importance sampling ratio, a component used in the evaluation, is almost zero. To fix this, KMIFQE learns a Gaussian kernel for the target policy and applies the kernel relaxation to the deterministic target policy, to avoid the zero in the importance sampling ratio.

**Strengths:**

- The paper clearly defines the research question, which is solving issues in the off-policy evaluation for a deterministic policy when applying in-sample learning.

- The theoretical part is sound. The paper provides the error bound of the value estimation and mathematically analyzes the bias and variance of the kernel relaxation. The optimal bandwidth of the kernel is also derived, for balancing the bias and variance. The theoretical results about the bias, variance, and optimal bandwidth are empirically checked as well.

- The experiment result section includes a visualization of the estimated Q value in a low-dimensional toy environment. This helps the reader to understand how the new method works, and I appreciate that.

**Weaknesses:**

I have questions about the significance of the paper and the experiment setting. Please see the questions below. I would be happy to change my score after our discussion.

**Questions:**

- I would appreciate it if the authors could explain more on the significance of the research question. The paper indicates the method is designed for continuous control tasks. In continuous control tasks, it is common to see an agent learn a stochastic policy instead of a deterministic policy, as many of the reinforcement learning algorithms suitable for continuous control tasks can work with a stochastic policy (BCQ, IQL, InAC, IAC, etc.), and it is also a simple and straightforward solution to avoid the zero importance sampling ratio. I think the importance of solving the issue in deterministic policy evaluation can be clearer if the paper can include an explanation of when people suffer from this issue in practice.

- Another question is about the experiment setting. When proposing the new method, the paper indicates that the method is for in-sample learning. However, in the experiment section, the target policy is trained by TD3. The behavior policy is different from the target policy and does not guarantee to be in-sample (the target policy is a stochastic policy generated from another TD3 learned policy, whose performance is 70%~80% of the target policy, according to the paper). It is not clear to me how the in-sample condition is ensured in the experiment.

---

> ### Author Response · Authors · 2023-11-15
> **Response to Reviewer $\text{\textcolor{blue}{bgFP}}$**
>
> Thank you for taking the time to provide valuable feedback on our work. Please let us know if any of our responses below need further clarification.
>
> $ $
>
> **Q1. The practicality of evaluating a deterministic policy and its example**
>
> Please note that our algorithm is not an offline RL algorithm that can choose the form of the target policy (stochastic/deterministic) but an algorithm for evaluating a given deterministic policy.
>
> We focused on evaluating deterministic target policies since there are many cases where the evaluation of deterministic policies is needed, e.g., safety-critical systems such as industrial robot control and drug prescription, where the sequences of actions need to be precisely and consistently controlled without introducing variability [Silver et al., 2014][Kallus and Zhou, 2018]. In these situations, one may want to evaluate the performance of the deterministic policies based on offline data when the robot control or drug prescription policy is updated before deployment.
>
> We included an explanation of the practicality of our problem setting and an example in the second paragraph of the introduction section.
>
>
> $ $
>
> [Silver et al., 2014] Silver et al. “Deterministic Policy Gradient Algorithms” ICML 2014.
>
> [Kallus and Zhou, 2018] Kallus and Zhou. "Policy Evaluation and Optimization with Continuous Treatments.." AISTATS 2018.
>
> $ $
>
> **Q2. How in-sample learning is conducted in the experiments?**
>
> Please note that in-sample learning does not mean on-policy learning. In-sample learning denotes the method that learns the value function by only using actions in the (off-policy) dataset [Xu et al., 2023][Kostrikov et al., 2022]. Even though the stochastic behavior policy and the deterministic target policy differ, it is in-sample learning if the actions used in the Q updates are in the dataset. Our method updates the Q-function with the update vector estimated only using the samples in the data following Eq.(6). Detailed procedure of our algorithm used in the experiments is presented in Algorithm 1 in Appendix B. Our algorithm works when the support of a deterministic target policy relaxed by a kernel is in the support of a stochastic behavior policy, and the experiment setting is made to satisfy the condition.
>
> To clear up how our algorithm does in-sample learning, we explicitly state in the main text that the pseudo-code of our algorithm is in Algorithm 1 in Appendix B. Furthermore, we modified Algorithm 1 to explain how the samples in the data are used to learn Q-function.
>
> $ $
>
> [Xu et al., 2023] Xu et al., "Offline RL with no OOD actions: In-sample learning via implicit value regularization." ICLR 2023.
>
> [Kostrikov et al., 2022] Kostrikov et al., "Offline reinforcement learning with implicit q-learning." ICLR 2022.

---

> > ### Comment · Reviewer_bgFP · 2023-11-16
> > **Reply**
> >
> > I would like to thank the author for a detailed explanation in their reply and the revised submission. After reading the response and other reviews, both the method and the contribution are more clear to me. I have increased my score.

---

> > > ### Author Response · Authors · 2023-11-16
> > > **Thank you for the quick response**
> > >
> > > Thank you for the quick response. We are glad to hear that your concerns are addressed. If you have further concerns, please let us know.

---

### Official Review · Reviewer_PFzG · 2023-10-31

**Soundness:** 3 good
**Presentation:** 2 fair
**Contribution:** 3 good
**Rating:** 8
**Confidence:** 3

**Summary:**

This paper proposes an extension of the in-sample TD learning algorithms of Schlegel et al and Zhang et al to the case where there is a deterministic target policy for continuous actions. Their extension modifies the deterministic policy to have support of a gaussian kernel centered on the action taken by the target policy (as opposed to a direct-delta function). They then derive the bias and variance of the new estimators, the mean squared error of the estimator, and other ancillary artifacts of the estimator.

**Strengths:**

Overall, I believe the paper is well positioned it the literature and the motivation behind the algorithm is stated clearly. While densely written, I believe the assumptions and proofs are reasonable, but I did not check each proof in-depth.

**Weaknesses:**

1. The paper is begging for an example of when you might want to evaluate a deterministic policy using data generated by a stochastic policy. From my experience, in reinforcement learning we often have to inverse problem. For instance, if using expert data generated through a pid controller or other classic control algorithm we will encounter deterministic behavior policies. Other than as baselines to compare RL algorithms, I’m struggling to understand when we might encounter the need to evaluate a deterministic policy. If this is the only example, that isn’t a deal breaker but an example would make the paper much better.
2. It is unclear what parts of the algorithm are learned and what is not learned/given. I believe A(s) is learned (in the full version of KMIFQE) and the value function it self is learned. Are any other parts learned? Maybe a section clearly stating the algorithm succinctly before or after proving the various artifacts would be beneficial to clarify exactly the moving parts.
3. There should be a statement on how the hyperparameters are chosen for all the methods. While this is in the appendix, there should be space made for this in the main paper. One notable missing part of this was a discussion on how to set the bandwidth. Is this given or determined by the data (through equation 10)? If it is given, are there any rules of thumb that would help practitioners? This is discusse briefly how this determines the bias-variance trade-off, but I believe this is lacking further exploration for practical use-cases.

**Suggestions:**

- Section 4/4.1 is written in an extremely dense manner. This is partially necessary due to the complexity of the math behind the kernel relaxation for the target policy, but I believe the authors could have done more to explain the theorems and propositions in more intuitive forms. I also believe separating section 4.2 into two sections could be beneficial (Optimal bandith/metric in one and bounding the contraction of the bellmen operators).

**Questions:**

See Above

---

> ### Author Response · Authors · 2023-11-15
> **Response to Reviewer $\text{\textcolor{green}{PFzG}}$**
>
> Thank you for taking the time to provide detailed feedback on our work. Please let us know if any of our responses below need further clarification.
>
> $ $
>
> **W1. Practicality of evaluating a deterministic policy using data generated by a stochastic policy and its example**
>
> There are many cases where the evaluation of deterministic policies is needed, e.g., safety-critical systems such as industrial robot control and drug prescription, where the sequences of actions need to be precisely and consistently controlled without introducing variability [Silver et al., 2014][Kallus and Zhou, 2018]. In these situations, one may want to evaluate the performance of the deterministic policies based on offline data when the robot control or drug prescription policy is updated before deployment. For the offline data, if the data is collected from the policies used in the past, the data can be viewed as sampled from a single stochastic behavior policy.
>
> We included an explanation of the practicality of our problem setting and an example in the second paragraph of the introduction section.
>
> $ $
>
> [Silver et al., 2014] Silver et al. “Deterministic Policy Gradient Algorithms” ICML 2014.
>
> [Kallus and Zhou, 2018] Kallus and Zhou. "Policy Evaluation and Optimization with Continuous Treatments.." AISTATS 2018.
>
> $ $
>
> **W2. Parts of the algorithm that are learned and not learned**
>
> We learn the kernel metric composed of $h$ (scale of the metric, referred to as bandwidth) and $A$ (shape of the metric, referred to as metric) as well as $Q$ function, using Eq. (10), Eq. (13), and Eq. (6) respectively. The pseudo-code of our algorithm is provided in Algorithm 1 in Appendix B, where we use the learned bandwidth $h$.
>
> We modified the paper's main text to guide readers to our pseudocode in Algorithm 1 in Appendix B and clear up what is learned and given in our algorithm. Furthermore, we modified Algorithm 1 to contain more details. We also modified the caption of Table 1 to explicitly mention what is learned in our algorithm in the experiment.
>
> $ $
>
> **W3. Hyperparameter selection**
>
> For the hyperparameters in our method, bandwidth $h$ in our method is learned by Eq. (10).  As for the IS ratio clipping range, we selected the range by grid search on Hopper-v2 domain from the cartesian product of minimum IS ratio clip values  {1e-5, 1e-3, 1e-1} and maximum IS ratio clip values  {1, 2, 10} and applied same hyperparameters on the other environments. For the hyperparameters in the baselines, we mostly followed the settings in SR-DICE as they experimented on the same or similar environments.
>
> We added a brief explanation of the hyperparameter settings in the experiment section and details in Appendix C.1, “Hyperparameters” section.
>
> $ $
>
> **S1. Intuitive explanation of the theorems and propositions**
>
> Thank you for your suggestion. We added intuitive explanations of the theorems and propositions in the main text. For Theorem1, we included an intuitive analysis of the leading-order bias in the paragraph after Theorem 1. For Proposition 3, we included an intuitive explanation of how the metric measures similarity between actions and how it is reflected in the importance resampling in the second paragraph after Proposition 3.
>
> $ $
>
> **S2. Separation of section 4.2 into Optimal bandith/metric section and error bound analysis section**
>
> Thank you for your suggestion. The contents related to the error bound are put in a separate subsection “4.3 Error Bound Analysis”

---

### Meta-Review · Area_Chair_BfBv · 2023-12-04

**Metareview:**

This paper studies the problem of off-policy policy evaluation in settings with continuous action space and deterministic target policy. For such a problem, the classic importance sampling approach would fail due to the importance sampling ratio being zero. The paper propose to relax the deterministic target policy to a stochastic one using kernel methods, and analyze the bias and variance resulting from such relaxation. Empirical evaluations are performed showing significantly improved accuracy comparing to existing baselines.

Overall this is a solid paper that proposes a novel method for an open problem of interest.

**Justification For Why Not Higher Score:**

The problem of evaluating deterministic policies, while applicable to some problems, is still relatively niche.

**Justification For Why Not Lower Score:**

NA

---

### Decision · Program_Chairs · 2024-01-16

Accept (spotlight)